

# Near-field emission profiling of Rainforest and Cerrado fires in Brazil during SAMBBA 2012

Amy K. Hodgson[1+], William T. Morgan[1], Sebastian O'Shea[1], Stéphane Bauguitte[2], Jams D. Allan[1,3], Eoghan Darbyshire[1], Michael J. Flynn[1], Dantong Liu[1], James Lee[4], Ben Johnson[5], Jim Haywood[6], Karla M. Longo[7*], Paulo E. Artaxo[8], and Hugh Coe[1]

[1]School of Earth & Environmental Sciences, University of Manchester, Manchester, UK
[2]Facility for Airborne Atmospheric Measurements, Cranfield University, UK
[3]National Centre for Atmospheric Science, University of Manchester, Manchester, UK
[4]Department of Chemistry, University of York, York, UK
[5]Met Office, Exeter, UK
[6]College of Engineering, Mathematics and Physical Sciences, University of Exeter, UK
[7]National Institute for Space Research (INPE), Sao Jose dos Campos, Brazil
[8]Physics Institute, University of Sao Paulo, Sao Paulo, Brazil
[+]Now at The Weather Company, Birmingham, UK
[*]Now at NASA Goddard Space Flight Center and USRA/GESTAR, Greenbelt, MD, USA

*Correspondence to:* W. T. Morgan
(will.morgan@manchester.ac.uk)

**Abstract.** We profile trace gas and particulate emissions from near-field airborne measurements of discrete smoke plumes in Brazil during the 2012 biomass burning season. The South American Biomass Burning Analysis (SAMBBA) Project conducted during September and October 2012 sampled across two distinct fire regimes prevalent in the Amazon Basin. Combined measurements from a Compact Time Of Flight Aerosol Mass Spectrometer (C-ToF-AMS) and a Single Particle Soot Photometer

(SP2) are reported for the first time in a tropical biomass burning environment. Emissions from a mostly-smouldering rainforest wildfire in Rondônia state and numerous smaller flaming Cerrado fires in Tocantins state are presented. While the Cerrado fires appear to be representative of typical fire conditions in the existing literature, the rainforest wildfire likely represents a more extreme example of biomass burning with a bias towards mostly-smouldering emissions. We determined fire integrated modified combustion efficiencies, emission ratios and emission factors for trace gas and particulate components for these two

fire types, alongside aerosol microphysical properties. Seven times more black carbon was emitted from the Cerrado fires per unit of fuel combustion ($EF_{BC}$ of $0.13 \pm 0.04$ g kg$^{-1}$) compared to the rainforest fire ($EF_{BC}$ of $0.019 \pm 0.006$ g kg$^{-1}$) and more than six times the amount of organic aerosol was emitted from the rainforest fire per unit of fuel combustion ($EF_{OC}$ of $5.00 \pm 1.58$ g kg$^{-1}$) compared to the Cerrado fires ($EF_{OC}$ of $0.82 \pm 0.26$ g kg$^{-1}$).

Particulate phase species emitted from the fires sampled are generally lower than those reported in previous studies and

in emission inventories, which is likely a combination of differences in fire combustion efficiency and fuel content, along with different measurement techniques. Previous modelling studies focussed on the biomass burning season in tropical South America have required significant scaling of emissions to reproduce in-situ and satellite aerosol concentrations over the region. Our results do not indicate that emission factors used in inventories are biased low, which could be one potential cause of the



reported underestimates in modelling studies. This study supplements and updates trace gas and particulate emission factors for fire type specific biomass burning in Brazil for use in weather and climate models. The study illustrates that initial fire conditions can result in substantial differences in terms of their emitted chemical components, which can potentially perturb the Earth system.

## 1 Introduction

Atmospheric aerosols represent the largest uncertainty in current understanding of radiative forcing of climate (Boucher et al., 2013), with biomass burning aerosol (BBA) aerosol-radiation interactions estimated to have a radiative forcing of $0.0 \ \mathrm{Wm}^{-2}$ but with a very large uncertainty of $\pm \ 0.2 \ \mathrm{Wm}^{-2}$ and significant perturbations on the regional scale (Boucher et al., 2013). BBA have both a global and regional effect on weather and climate via perturbation of the atmospheric radiation balance and cloud microphysical properties (Ramanathan et al., 2001; Andreae et al., 2004) and visibility (Andreae et al., 1988) but can also affect human health (Cançado et al., 2006; Arbex et al., 2007; Carmo et al., 2013). Biomass burning in the tropics contributes to more than 80% of all the emissions produced from total biomass burning globally (Ward et al., 1992). The Amazon Basin in Brazil contains approximately $4\mathrm{x}10^6\mathrm{km}^2$ of evergreen tropical forest (Christian et al., 2007) and during the dry season (August-October) intense widespread burning occurs leading to high levels of atmospheric particulate matter (Chand et al., 2006). Both tropical rainforest fires and savannah-like fires occur in Brazil (Yamasoe et al., 2000) for land clearing and pasture maintenance (Martin et al., 2010), which leads to high levels of black carbon, organic matter and gas phase species in the atmosphere. Detailed representation of the emissions and properties of gas and particulate phase species from BB in Brazil are therefore required in global climate models for their outputs to be accurate and reliable (Bowman et al., 2009).

Emissions from BB are quantified either by emission ratios (ERs or $\mathrm{ER}_{x/y}$, the relative excess amounts of two smoke species, $x$ and $y$) or emission factors (EFs, grams of species released per kg of dry fuel burnt) and these are used in order to compute trace gas and particulates released from biomass burning fires. Numerous scientific studies have taken place to study smoke from biomass burning both in the field (e.g. Reid and Hobbs, 1998) and in the laboratory (e.g. McMeeking et al., 2009). Every fire is unique, differing in vegetation type and combustion stage e.g. flaming or smouldering, while other factors such as moisture content of the fuel, the environmental conditions and whether the vegetation is dead or alive can alter the emissions of certain BB species (e.g. Ward et al., 1996; Yokelson et al., 1996). However, while there is significant inter-fire variability, fires over a particular region often exist within broader distinct regimes depending on the dominant fuel type and combustion properties. Recent studies by McMeeking et al. (2009) and Akagi et al. (2011) have compiled emissions from different vegetation types and showed large variations in the composition of the emitted species and their overall abundance. All of these previous studies provide data for emission inventories which can then be directly used in atmospheric models. However, due to the large uncertainties and factors influencing BB emissions, further understanding of these emission variations are needed.

Recent instrument developments including the Time of Flight Aerosol Mass Spectrometer (ToF-AMS, Aerodyne Research, Inc., Billerica, MA, USA, Canagaratna et al., 2007) and a Single Particle Soot Photometer (SP2, Droplet Measurement Technologies, Boulder, CO, USA, Stephens et al., 2003) have been used recently to study biomass burning emissions in boreal





regions (e.g. Kondo et al., 2011; Jolleys et al., 2015) but previous measurements of particulate emissions from tropical biomass burning were conducted over a decade ago using instruments with low time resolution and sensitivity for particulate species, which do not allow single plumes to be characterised. This is particularly true for measurements from aircraft platforms and since ground and airborne sampling may sample smouldering and flaming contributions differently it is important to compare

such studies. The South American Biomass Burning Analysis Project (SAMBBA) is the first experiment to utilise both of these instruments in a tropical environment for studies of biomass burning. These fire type specific measurements are important as recent biomass burning studies in Brazil have found a shift from forest to savannah burning, which will impact trace gas and particulate emissions in the region (Ten Hoeve et al., 2012; Chen et al., 2013).

The airborne campaign of SAMBBA conducted twenty flights from 13 September to 4 October 2012. Two of the flights

focused on near-source biomass burning emissions, sampling across contrasting environments in terms of vegetation and fire regime. We present emission ratios and emission factors for a range of gas and particle phase species, alongside measurements of the physical and chemical properties of the plumes sampled by the aircraft.

## 2  Experimental Details & Methodology

### 2.1  Instrumentation

In-situ measurements presented here took place on the UK Facility for Airborne Atmospheric Measurement (FAAM) Airborne Research Aircraft, BAe-146. The BAe 146 aircraft has a wide range of instruments on board, but only those relevant to the analysis are discussed below. Mass concentrations of particulate species are reported at standard temperature and pressure in $\mu g \, sm^{-3}$ (where $sm^{-3}$ refers to standard cubic metre at 273.15 K and 1013.25 hPa).

A compact variant of the ToF-AMS provided real time size-resolved mass measurements of non-refractory (NR) organic

aerosol (OA) and inorganic component mass: sulphate, nitrate, chloride and ammonium (Drewnick et al., 2005; Canagaratna et al., 2007). This instrument provides quantitative high time resolution data with high precision and accuracy making it ideal for use on aircraft campaigns. Measured mass concentrations for the C-ToF-AMS have an uncertainty of approximately 30% (Bahreini et al., 2009). Previous studies by Crosier et al. (2007), Morgan et al. (2009) and Morgan et al. (2010) have detailed the sampling strategy and calibration protocols for the AMS on the BAe 146. Plume interceptions utilised the 'fast mass spectrum'

mode of the AMS (Kimmel et al., 2011), which provided data at 1 second time resolution. The instrument was calibrated using monodisperse ammonium nitrate and ammonium sulphate to provide the ionisation efficiency of nitrate, along with relative ionisation efficiencies for sulphate and ammonium. A collection efficiency of 1.0 was applied to the data based on comparisons with a Scanning Mobility Particle Sizer (SMPS) using data from the entire campaign (further details available in Allan et al., 2014). This is supported by the independent measurements of Brito et al. (2014) who also reported a collection efficiency of

1.0 for their Aerosol Chemical Speciation Monitor (ACSM) measurements conducted at a ground-site in Porto Velho during the SAMBBA experiment.

The early flights of the campaign (up to and including B737 on 20 September 2012) suffered from a partial blockage of the AMS pinhole (part of the inlet system where particles enter the instrument aerodynamic lens), which reduced the reported mass





concentrations. When comparing the AMS concentrations with optical particle counter and total scattering measurements, a clear and consistent discrepancy was evident pre- and post-blockage. In order to correct for this, a scaling factor was applied to recover the mass concentrations by comparing the mass concentrations to total scattering coefficients measured by a TSI nephelometer. The scaling factor was calculated using data from low-altitude straight and level runs within the boundary layer

when the sample line humidity was below 40%. The applied scaling factor was 2.69 and is applied to the data for B737 in this study.

The SP2 provides a determination of the single particle BC mass, the number of particles containing BC and the total mass of particles containing refractory black carbon (rBC) species (Baumgardner et al., 2004; Schwarz et al., 2006). The term rBC is defined as the incandescent material measured by the SP2, following the definition of Petzold et al. (2013). The SP2

instrument operation and subsequent data interpretation have been described elsewhere (Liu et al., 2010; McMeeking et al., 2010). Calibration of the SP2 incandescence signal in order to calculate single particle rBC mass was accomplished using monodisperse Aquadag BC particle standards (Aqueous Deflocculated Acheson Graphite, manufactured by Acheson Inc., USA) using a scaling factor of 0.75 (Baumgardner et al., 2012). A 30% uncertainty in the SP2 black carbon mass is used as in previous studies (e.g. McMeeking et al., 2010, 2012). During near-source plume sampling, multiple coincident particles may

be sampled at the same time by the instrument and such peaks in each single particle event are identified by the data analysis software, with the mass loading being the summation of the single particle masses from the identified peak signals.

The C-ToF-AMS and SP2 both sampled via Rosemount inlets (Foltescu et al., 1995). These have been shown to enhance aerosol concentrations dependent on the mean bulk density of the aerosol sample (Trembath et al., 2012), with the enhancement being largest for the super-micron size range (e.g. up to a factor of 10 for Saharan dust). For European pollution aerosol, which

has a comparable density to BBA, the enhancement is negligible for particles below an optical diameter of 0.6 $\mu m$. Given the size ranges of the C-ToF-AMS and SP2 and the general dominance of sub-micron aerosol in this environment based on size distribution measurements, limited enhancement is expected for the measurements presented here.

The C-ToF-AMS and SP2 measured downstream of nafion driers to prevent condensation of water in the inlet lines, which combined with the cabin temperature exceeding the ambient temperature, resulted in the sample being dried to a significant

extent. Sample line relative humidity measurements were typically between 20-60% during flights in Rondônia and from 20-30% in Tocantins at the flight altitudes of interest here.

The Fast Greenhouse Gas Analyzer (FGGA, Model RMT-200, Los Gatos Research Ltd., USA) utilises a cavity-enhanced absorption spectrometer to provide high accuracy, 1 Hz measurements of carbon dioxide and methane mixing ratios with a 0.1% uncertainty (O'Shea et al., 2013) and the VUV Fast Fluorescence CO Analyser measures carbon monoxide mixing ratios

with a 2% uncertainty (Hopkins et al., 2006; O'Shea et al., 2013).





## 2.2 Biomass burning emission calculations

### 2.2.1 Background ambient and in-plume measurements

Excess mixing ratios of species $x$ ($\Delta x$) are needed in order to calculate the ER and EF of a species. In order to calculate $\Delta x$, the ambient background mixing ratios of species $x$ must be subtracted from the values measured in the smoke plume. The ambient

background mixing ratio was defined as the fifth percentile for each species while outside the plume during constant-altitude runs in the boundary layer for each flight respectively. Plume identification was performed manually based on the time series of CO, OA and rBC. We note the discussion by Yokelson et al. (2013) that examines the limitations of the excess mixing ratios approach due to changes in background air composition through tropospheric mixing; as our measurements were made close to initial source through numerous plume intercepts on both flights and background concentrations were constant throughout,

we do not consider this mixing of background air into the plume to be a significant effect in this study.

The numerous instruments on-board the BAe-146 each had different response times and inlet lag times leading to difficulties when comparing data from different instruments. Therefore, an integral based approach was used which helps to compensate for these different response times (Yokelson et al., 1996; Karl et al., 2007; Yokelson et al., 2009, 2011). Integrated methods have been found to be more robust and decrease uncertainty compared to the direct point by point method (Karl et al., 2007).

Given the AMS measures Organic Matter (OM), rather than Organic Carbon (OC), which is the most typical reported value for OA in the literature and emission inventories, OM/OC is converted using a value of 1.6 following the work of Yokelson et al. (2009) and Akagi et al. (2012) for fresh biomass burning. The OM/OC ratio value is composition, source and age dependent, with values ranging from 1.4 for fresh urban aerosol to 2.2 for aged non-urban aerosol (Turpin and Lim, 2001), therefore this adds another source of uncertainty to the calculated OC emissions.

### 2.2.2 Modified combustion efficiency

The combustion efficiency (CE) and modified combustion efficiency (MCE) can be used to define the relative amount of flaming or smouldering combustion taking place. The CE is defined as the ratio of carbon emitted as $CO_2$ to the total carbon emitted. Total carbon emitted includes $CO_2$, CO, $CH_4$, hydrocarbons and carbon containing particulates (Ward and Radke, 1993). Measuring all of these emitted carbon species can be difficult in field campaigns, therefore we use the MCE which is

25 defined below following Ward and Radke (1993):

$$MCE = \frac{\Delta CO_2}{\Delta CO_2 + \Delta CO} \tag{1}$$

CE and MCE are closely related with a difference of only a few percent, as CO and $CO_2$ represent the majority of the carbon species emitted (Ward and Radke, 1993; Ferek et al., 1998). MCE can be interpreted as relative scale of varying degress of smouldering and flaming combustion; values greater than 90% are typically biased towards a fire in the flaming stage,

whereas a MCE less than 90% is defined as being biased towards the smouldering phase (Ward and Radke, 1993). The excess concentrations of CO and $CO_2$ were integrated over the plume interception time to give integrated excess values. These values for each plume interception were then plotted with the intercept forced to zero to give the fire average MCE.





### 2.2.3 Emission ratio

The absolute concentration of trace gases and particulates in fire plumes cannot directly be used to interpret emissions due to the dilution of the species with the ambient background air. This is particularly important when sampling smoke from aircraft platforms. Therefore ERs are calculated to give the relative emission of species $x$ to a simultaneously measured reference gas,

usually CO or $CO_2$ as these gases are non-reactive and conserved (Andreae and Merlet, 2001; Sinha et al., 2003). The ER of species $x$ using CO as the reference gas is defined below:

$$ER_x = \frac{\Delta x}{\Delta CO} \tag{2}$$

For gas phase species, $ER_x$ is usually given as the molar ratio and for aerosol species ER is stated as the mass ratio at 273.15 K and 1013.25 hPa. When only one pass is made through a BB plume the calculation of $ER_x$ is trivial using the equation

above. The $ER_x$ can also be derived when multiple passes are made through the BB plume by using the regression slope of the excess species concentration of $x$ versus the reference species with the line forced through zero (Yokelson et al., 1999). This is the method chosen in this study, where the excess concentrations of species $x$ and the reference species concentration are integrated over the plume intercept, with the regression slope of these species giving $ER_x$.

### 2.2.4 Emission factor

Another parameter used to define the emission of a particulate species from fires is the EF. EF is reported as the mass of species $x$ emitted per kg of dry fuel burnt. The dry fuel burnt is approximated by the total mass of carbon species released in the form of $CO_2$, CO, $CH_4$, non-methane hydrocarbons, particulate carbon etc (Yokelson et al., 2013; Stockwell et al., 2015). EFs were calculated using the carbon mass balance method (Ward and Radke, 1993; Yokelson et al., 1996). The mass fraction of carbon in the fuel is needed for EF calculation and as this quantity was not measured during the campaign we used a value of 0.5

$\pm 0.05$, which is typical in the literature (e.g. Wooster et al., 2011). Susott et al. (1996) presented data that shows the carbon content of Brazilian vegetation ranges from 45-55%. As we only used $CO_2$, CO, $CH_4$ as the carbon containing species in the EF calculations, our EFs are likely to be overestimated by 1-2% (Susott et al., 1996; Andreae and Merlet, 2001), although the particulate phase carbon is usually only a small fraction of the carbon emitted (Lipsky and Robinson, 2006; McMeeking et al., 2009). The EF for species $x$ ($g\,kg^{-1}$) is defined below following Yokelson et al. (1999) and Wooster et al. (2011):

$$EF_x = F_c 1000 \frac{MM_x}{MM_{carbon}} \frac{C_x}{C_T} \tag{3}$$

Where $1000\,g\,kg^{-1}$ is a unit conversion factor, $MM_x$ is the molecular mass of species $x(g)$, $MM_{carbon}$ is the molecular mass of carbon (12) and $C_x/C_T$ is the ratio of the number of moles of species $x$ in the plume interception to the total number of moles of carbon, which is calculated following Yokelson et al. (2009) and Wooster et al. (2011):

$$\frac{C_x}{C_T} = \frac{ER_{x/CO_2}}{\sum_{j=1}^{n}(NC_j ER_{j/CO_2})} \tag{4}$$

Where $ER_{x/CO_2}$ is the ER of species $x$ to $CO_2$, $NC_j$ is the number of carbon atoms in compound $j$, and the sum is over all carbon species including $CO_2$ (e.g. Wooster et al., 2011).



Uncertainties in the EFs are derived in quadrature from the uncertainty in the carbon content of the fuel (0.05) and the uncertainty in the associated ER values.

## 3 Results

### 3.1 Flight Overview

Two of the flights during SAMBBA focussed on near-field in-situ measurements of active fires. The fires were sampled within the boundary layer, with out-of-plume aerosol samples dominated by biomass burning haze. The general features of the fires are summarised below.

### 3.1.1 Rondônia flight

Flight B737 took place in Rondônia State in the West of Brazil on 20 September 2012, with take-off at 14:45 UTC (10:45 local time) and a duration of 3 hours 45 minutes. The natural vegetation in Rondônia is characterised by dense Amazonian rainforest, but the region has become one of the most deforested areas of the Amazon. Fig. 1a shows a large smouldering rainforest fire, which was suspected to be a natural wildfire, likely initiated by lightning. The fire was located in a National Park many kilometres from the nearest road, in a region well away from any deforestation. It is therefore unlikely the fire was a deforestation fire, which is the dominant form of fire in the region and the typical focus of previous campaigns. MODIS hotspot data from the TERRA overpass at 14:26 UTC on 19 September 2012 indicated that this fire was likely started that day. The near field plume interceptions shown on the flight track for B737 on Fig. 1b took place at an altitude of 1800m (above sea-level) with far field interceptions at an altitude of 2500m (above sea-level). The fire was located on a 900m high plateau, therefore the plume was intercepted at 900m above the fire, with smoke estimated to be approximately 6 minutes old (based on vertical wind velocity measurements). This paper only focuses on the near-field measurements to understand initial emissions, while a future publication (Morgan et al., in prep) will characterise the ageing and transformation of the plume downwind.

Some mid-level cloud was present in central Rondônia and a large pyro-cumulus cloud was observed over the BB plume above the boundary layer. Winds were from the North-North-West and relative humidity outside of plume interceptions was high with values of around 70% at 900m. During this flight, 9 separate plume interceptions took place, each lasting approximately 15 seconds, shown in Fig. 1c with large increases in CO, rBC and OA clearly visible. Plume interceptions were made prior to the pyrocumulus cloud. Background concentrations of CO, OA and rBC were 213 ppbv, 9.81 $\mu$gsm$^{-3}$ and 0.31 $\mu$gsm$^{-3}$ respectively. Plume maxima ranged between 1261-29554 ppbv for CO, 134-3661 $\mu$gsm$^{-3}$ for OA and 1-9 $\mu$gsm$^{-3}$ for rBC.

The Rondônia fire MCE of 0.79 $\pm$ 0.02 is effectively identical to the MCE of 0.788 for residual smouldering combustion of logs in Brazil from a ground-based experiment (Christian et al., 2007), although compared to other rainforest-like fires reported in the literature that are summarised in Table 1, the MCE for Rondônia fire is much lower e.g. Ferek et al. (1998) reported a value of 0.87 for smouldering forest fires in Brazil measured on an aircraft.





### 3.1.2 Tocantins flight

Flight B742 took place in the Tocantins State of Brazil on 27 September 2012, with take-off at 13:00 UTC (10:00 local) and a flight duration of 3 hours 15 minutes. Tocantins State is characterized by Cerrado vegetation, in particular grasslands (campo limpo/campo sujo) and open woodland (Cerrado sensu) forms (Mistry, 1998). Fig. 1a shows an example of some of the fires
sampled during the Tocantins flight with flames visible in the closest fire. The vegetation consists mainly of grassland with some trees. During the flight, numerous new fires were starting, which are likely a consequence of man-made agricultural burning based on existing knowledge of fire in the region (e.g. Longo et al., 2013). The BB smoke plumes were sampled at an altitude of 600m above the fires, with smoke sampled being approximately 4 minutes old, which we define as initial smoke. The flight track is shown in 1b, with the MODIS hotspot data from NASA's Terra satellite (Kaufman et al., 1998, 2003; Giglio
et al., 2006), shown by the red markers to indicate the fire locations and the plume interceptions shown by the blue markers.

    There was little cloud cover in the area, with low relative humidity values of around 30% at an altitude of 600m outside of plume interceptions and winds were light coming from the South East at 950 hPa. During this flight, 23 plume interceptions took place each lasting between 5 to 10 seconds. The plume interceptions can clearly be seen in the time series of CO, rBC and OA shown in Fig. 1c. Background concentrations at 600m altitude were 228 ppbv for CO, 0.77 $\mu$gsm$^{-3}$ for rBC and 9.31
$\mu$gsm$^{-3}$ for OA. Maximum concentrations in the plume interceptions ranged between 750-17732 ppbv for CO, 10-110$\mu$gsm$^{-3}$ for rBC and 65-1636 $\mu$gsm$^{-3}$ for OA.

    The MCE of the Tocantins fires was $0.94 \pm 0.02$, identical to similar aircraft measurements of Cerrado fire emissions reported by Ferek et al. (1998). Compared with the existing literature on savannah/Cerrado fires (see Table 1), the Tocantins MCE is very similar e.g. African savannah fires (Yokelson et al., 2003) and California chaparral fires (Akagi et al., 2012) have
similar MCE values of 0.94 and 0.93 respectively.

### 3.2 Trace gas emissions

Fig. 2 shows the scatter plots used for derivation of the trace gas ERs, with the derived values shown in Table 2, with an uncertainty of one standard deviation in the line of best fit. The trace gas species measured on the aircraft are very strongly correlated, with r-squared values between 0.92 and 0.99 illustrating the common source of these species i.e. active fires. The
different points in the Tocantins figures are derived from data from multiple fires and the lack of variability indicates similarity between the fires, although the level of emission does vary significantly. Table 3 shows the calculated EF values for $CO_2$, CO and $CH_4$. Also presented are reported values from other studies from the literature.

    The Rondônia fire $EF_{CO_2}$ of $1447 \pm 145$ g kg$^{-1}$ and $EF_{CO}$ of $237 \pm 24$ g kg$^{-1}$ are very similar to those reported by Christian et al. (2007) for smouldering logs in Brazil, which were 1346 g kg$^{-1}$ and 229 g kg$^{-1}$ respectively. Our $EF_{CO_2}$ is
similar to previous studies reporting emission factors for deforestation fires in Brazil e.g. Ward et al. (1992), Kaufman et al. (1992), Ferek et al. (1998) and Yokelson et al. (2007). However, our $EF_{CO}$ is typically 2-3 times larger than other previous studies in Brazil, other than the Christian et al. (2007) study. Global average values reported in the literature are typically slightly larger in the case of $EF_{CO_2}$ and significantly lower in the case of $EF_{CO}$. Andreae and Merlet (2001) and Akagi et al.





(2011) determined tropical forest global averages that were 11% and 14% larger than our reported $EF_{CO_2}$, while the GFEDv3/4 and GFASv1.0 emission inventories (van der Werf et al., 2010; Kaiser et al., 2012) report a value that is 12-14% greater; $EF_{CO}$ values are 56%, 61% and 57-61% lower.

Our value of $5.17 \pm 0.53 \, \mathrm{g \, kg^{-1}}$ for $EF_{CH_4}$ is 2-3 times lower than those reported by Christian et al. (2007) and Ferek et al. (1998) but similar to those reported by Kaufman et al. (1992) and Yokelson et al. (2007). The global averages for tropical forest also agree well ($5.07 \pm 1.98 \, \mathrm{g \, kg^{-1}}$ (Akagi et al., 2011) and $6.8 \pm 2.0 \, \mathrm{g \, kg^{-1}}$ (Andreae and Merlet, 2001)). The value of 6.6 $\mathrm{g \, kg^{-1}}$ used in the GFEDv3 and GFASv1.0 emission inventories for deforestation fires is 28% higher than our reported value for the Rondônia tropical forest fire, although the latest GFEDv4 release reports a value of $5.07 \, \mathrm{g \, kg^{-1}}$, which is very similar to ours.

Emission factors for $CO_2$ and CO for the Tocantins were $1711 \pm 175 \, \mathrm{g \, kg^{-1}}$ and $74 \pm 8 \, \mathrm{g \, kg^{-1}}$, which are similar to existing values reported in the literature for Cerrado fires in Brazil. Similarly, our observed $EF_{CO_2}$ is comparable to global average savannah and grassland values from Andreae and Merlet (2001) and Akagi et al. (2011), which are 6% and 1% lower respectively, while being 4% higher than the value of $1646 \, \mathrm{g \, kg^{-1}}$ used in the GFEDv3 and GFASv1.0 emission inventories for savannah fires. For $EF_{CO}$, our value is 14%, 17%, 21% and 17% larger than those reported by Andreae and Merlet (2001), Akagi et al. (2011), GFEDv3/GFASv1.0 and GFEDv4 emission inventories respectively. The Tocantins fires $EF_{CH_4}$ value of $2.23 \pm 0.23 \, \mathrm{g \, kg^{-1}}$ is similar to previous measurements of Cerrado fires in Brazil and global average savannah fires.

### 3.3 Particulate emissions

The total excess mass of aerosol species integrated over the plume interceptions as a percentage of the total mass of aerosols measured (BC, OA, chloride, ammonium, sulphate and nitrate) are presented in Fig. 3 and Table 4.

The aerosol emitted by the Rondônia fire was composed of over 97% organic mass, greater than the value of 86.55% reported by Ferek et al. (1998) for a smouldering Brazilian rainforest fire. The Ferek et al. (1998) value of 7.75% for BC mass is over an order of magnitude greater than our value for smouldering rainforest BC mass of 0.3%.

The value of 88.12% for OM for Cerrado fires in Brazil reported by Ferek et al. (1998) is similar to our value of 84.4% for the Tocantins Cerrado fires. Ferek et al. (1998) values for BC (6.76%), chloride (2.98%) and nitrate (1.29%) for Cerrado fires are also all comparable to our measurements for Brazilian Cerrado of 7.99%, 5.08% and 1.29% respectively.

The Rondônia fire emitted 12.7% more OA than the Tocantins fire in terms of their average mass fraction. The flaming Cerrado fires emitted over twenty five times more BC by mass to the total particulate mass than the smouldering rainforest fire. Yamasoe et al. (2000) found the difference was only three times as much when comparing tropical deforestation and cerrado fires in Brazil. The Tocantins fires emitted almost ten times more Cl- by mass of total particulates than the Rondônia fire, which is similar to Yamasoe et al. (2000) who found the difference was approximately eleven times more for the flaming Cerrado fires compared to the smouldering rainforest fire. Grass is known to be high in chlorine (Lobert et al., 1999), which would explain the relative abundance of chloride sampled from the Tocantins fires.

Fig. 4 shows the black carbon mass and number size distributions for the Rondônia and Tocantins fires which gives an indication of the size of particles at source. The grey shading shows the minimum and maximum size distributions from




the plume intercepts on each flight to show the plume to plume variations, with the solid black line indicating the average mean. There is little difference in the BC size distributions of the two fires despite the large difference in fuel and burning characteristics. Average mass median diameters are 0.19 μm and 0.20 μm for B737 and B742 respectively, while average number median diameters are 0.10 μm for both fires, calculated from the log-normal fits based on the distributions shown in Fig. 4.

Fig. 2 includes the scatter plots used to derive the particulate species ERs, while their derived values are listed in Table 5. Particulate phase species and trace gas emissions are strongly correlated, with r-squared values of between 0.72 and 0.98 illustrating their common sources. Table 6 shows the calculated EF values for particulate species with their associated uncertainties, alongside reported values from the literature for comparison. Given that directly comparable measurements from fires in Brazil are more scarce for particulate emissions than trace gases and the substantial range from region-to-region reported in the literature (e.g. Jolleys et al., 2012), in the following text we focus on comparing with global average values and those used in emission inventories in the absence of Brazil-specific emission factors.

### 3.3.1 Organic aerosol

The Rondônia and Tocantins fire $EF_{OC}$ values were $5.00 \pm 1.58$ g kg$^{-1}$ and $0.82 \pm 0.26$ g kg$^{-1}$ respectively, representing more than a six-fold increase in OC per kg fuel burnt when comparing the two fires.

The Rondônia fire $EF_{OC}$ is 3.5 times lower than the smouldering forest value of $17.9 \pm 7.6$ g kg$^{-1}$ reported by Ferek et al. (1998). From a global perspective, our value is very similar to those reported by Akagi et al. (2011) and Andreae and Merlet (2001) for average global tropical forests, which were $4.71 \pm 2.73$ g kg$^{-1}$ and $5.2 \pm 1.5$ g kg$^{-1}$ respectively. Our value is 16% higher than the $EF_{OC}$ value of 4.3 g kg$^{-1}$ used for deforestation fires in the GFEDv3 and GFASv1.0 emission inventories.

Ferek et al. (1998) reported a $EF_{OC}$ of $5.9 \pm 2.8$ g kg$^{-1}$ for Cerrado burning, which is over seven times greater than our $EF_{OC}$ for the Tocantins fire. Global averaged savannah values for $EF_{OC}$ reported by Akagi et al. (2011) and Andreae and Merlet (2001), which were $2.62 \pm 1.24$ g kg$^{-1}$ and $3.4 \pm 1.4$ g kg$^{-1}$ respectively, are 3-4 times greater than our value. Similarly, the GFEDv3 and GFASv1.0 emission inventories value of 3.2 g kg$^{-1}$ for savannah fires is almost four times higher than the value we calculated for the Tocantins fire.

### 3.3.2 Black carbon

Our $EF_{BC}$ values were $0.019 \pm 0.006$ g kg$^{-1}$ and $0.13 \pm 0.04$ g kg$^{-1}$ for the Rondônia and Tocantins fires respectively, approximately one order of magnitude apart.

Our $EF_{BC}$ value for the Rondônia fire is close to two orders of magnitude smaller than the value of $1.5 \pm 0.9$ g kg$^{-1}$ reported by Ferek et al. (1998) for smouldering rainforest fires in Brazil. This divergence between our $EF_{BC}$ value and those in the literature is similar when comparing with global averages; values of $0.52 \pm 0.28$ g kg$^{-1}$ (Akagi et al., 2011) and $0.66 \pm 0.31$ g kg$^{-1}$ (Andreae and Merlet, 2001) for global average tropical forest fires are more than an order of magnitude greater than our reported value. The GFEDv3 and GFASv1.0 emission inventories use a value of 0.57 g kg$^{-1}$ for deforestation fires,





which is again over an order of magnitude greater than our value for the Rondônia fire. Our value is similar to those measured for smouldering Indonesian peat by Stockwell et al. (2016).

Compared with Ferek et al. (1998), our $EF_{BC}$ is more than a factor of 5 smaller than their value of $0.7 \pm 0.4 \, \mathrm{g \, kg^{-1}}$ for Cerrado fires in Brazil. $EF_{BC}$ values for African savannah fires and global average savannah fires are also larger than our value, $0.39 \pm 0.19 \, \mathrm{g \, kg^{-1}}$ (Sinha et al., 2003) and $0.37 \pm 0.20 \, \mathrm{g \, kg^{-1}}$ (Akagi et al., 2011) respectively. The GFEDv3 and GFASv1.0 emission inventories use a value $0.46 \, \mathrm{g \, kg^{-1}}$, which is 3.5 times greater than the value we calculated for the Tocantins fires.

### 3.3.3 Inorganic aerosol

Values for $EF_{Cl}$ of $0.04 \pm 0.01 \, \mathrm{g \, kg^{-1}}$ and $0.09 \pm 0.03 \, \mathrm{g \, kg^{-1}}$ for the Rondônia and Tocatins fires respectively and are 2-4 times smaller than global averages reported by Akagi et al. (2011) for tropical forest and savannah fires. For the Rondônia and Tocantins fires, values for $EF_{NO_3}$ were $0.078 \pm 0.025 \, \mathrm{g \, kg^{-1}}$ and $0.013 \pm 0.004 \, \mathrm{g \, kg^{-1}}$; global average tropical forest and savannah fires reported in Akagi et al. (2011) are 41% and 23% greater than our reported values respectively. The Rondônia fire $EF_{SO_4}$ value of $0.034 \pm 0.011 \, \mathrm{g \, kg^{-1}}$ is close to a factor of four smaller than the value of $0.133 \, \mathrm{g \, kg^{-1}}$ for the global tropical forest average reported in Akagi et al. (2011). The Tocantins fire $EF_{SO_4}$ value of $0.0006 \pm 0.0002 \, \mathrm{g \, kg^{-1}}$ is thirty times smaller than the value of $0.018 \, \mathrm{g \, kg^{-1}}$ for the global average of savannah fires in Akagi et al. (2011). Values for $EF_{NH_4}$ were $0.033 \pm 0.011 \, \mathrm{g \, kg^{-1}}$ and $0.015 \pm 0.005 \, \mathrm{g \, kg^{-1}}$ for the Rondônia and Tocantins fires respectively, which are approximately six and four times greater than the global averaged tropical forest and savannah fires reported by Akagi et al. (2011).

## 4 Discussion

### 4.1 How representative are the Rondônia and Tocantins fires?

Section 3.2 reports comparisons between our gas phase emission factors and those in the existing literature and emission inventories, which can serve as a basis for judging the representativeness of our particle phase measurements. Gas phase emission factors are more numerous, up-to-date and robust than their particle phase counterparts, so we focus on those to place our measurements in the context of the existing literature.

For the major trace gas emissions reported here, the Tocantins fire emission factors are very similar to previous measurements in the Brazilian Cerrado as well as global average savannah and grassland fires; this suggests from a gas-phase perspective, the Tocantins fires are consistent with previous measurements and likely representative of typical flaming Cerrado fires. Flaming combustion is predominant in Cerrado fires due to the dry fine fuel, which burns quickly with high combustion efficiency (generally of 0.93 or greater e.g. Ward et al. (1992); Ferek et al. (1998)), which is consistent with our observations of the Tocantins fires.

For the Rondônia wildfire, our $EF_{CO_2}$ value is similar to previous emission factors reported for deforestation fires in Brazil, as well as global average values and those used in emission inventories. However, our value for $EF_{CO}$ is 2-3 times greater than those reported in previous studies aside from one study by Christian et al. (2007) focussing on smouldering logs in





Brazil. For $EF_{CH_4}$, there is no clear discrepancy between our value and those reported across the literature, although the range of values is large for deforestation and tropical fires. This suggests that the Rondônia fire represents a mostly-smouldering example of biomass burning in Brazil; deforestation fires in Brazil have been shown to have a more balanaced mix of flaming and smouldering combustion e.g. Ward et al. (1992) observed combustion efficiencies ranging from 0.88 to less than 0.80. A

further factor that will lower the combustion efficiency is the water content of the fuel, which was likely much greater for the Rondônia fire as a growing forest, whereas typical deforestation fires may have seen the fuel dried for a season before burning. A key outstanding question is how different stages of combustion evolve for these types of fires, with our Rondônia example likely representing one extreme of this evolution as a mature wildfire.

## 4.2   Particulate emissions compared to existing literature

For the Rondônia fire, organic aerosol made up 97% of the emitted particulate mass on average, which is very similar to global average values for tropical forests and those used for deforestation fires in emission inventories. Values of $EF_{OC}$ are scarce in the literature for Brazilian biomass burning fires, with the only other comparable value from Ferek et al. (1998) being 3.5 times greater than our value. Major differences are found between our emission factors for the other particulate species measured by our study when compared with existing literature, particularly in the case of rBC, which was more than an order of magnitude

smaller. These differences are likely due to the mostly smouldering nature of the fire, with $EF_{BC}$ being strongly coupled to combustion efficiency and the fuel type of the vegetation. rBC made up just 0.3% of the emitted mass of particulate species, which is much less than that observed at the regional scale during SAMBBA (5.5-6.1%) as reported by Darbyshire et al. (in prep.). In order for conditions during the Rondônia fire to be typical of biomass burning emissions during the study, substantial evaporation or loss of non-rBC aerosol species would be required, which is not observed when assessing transformations at the

plume or regional scale (Morgan et al., in prep.). Therefore we conclude that the conditions prevalent during the Rondônia fire are unlikely to represent the dominant mode of biomass burning emissions during the wider study.

While particulate emission factors for the Tocantins fire were generally of the same order of magnitude to values for average global savannah and grassland emissions, they were lower by factors of 2-4. Emission factors for inorganic particulate species from these environments are severely lacking, so drawing specific conclusions would be unwise. For OC and BC,

more emission factors are available in the literature, so consideration of these differences is more warranted. Given that the combustion efficiency of the fires were very similar to those previously reported in the literature, the most likely candidates for the differences are the fuel type and sampling methods.

Assessing the role of fuel type is not possible within this study but we can discuss potential biases due to sampling methods. Previous measurements have relied on a variety of methods involving prior collection on a filter followed by thermal, optical

or combined thermal-optical techniques, which are then either analysed off-line or in real time. Thermal-based approaches are prone to biases due to pyrolysis or charring of carbonaceous material during the analytical protocol, which can make separation of the OC and BC components challenging and uncertain (e.g. Chow et al., 2007; Petzold et al., 2013). Optical measurements can overestimate BC due the presence of other absorbers such as OA as well as optical interactions between particles and the filter matrix (Bond and Bergstrom, 2006; Lack et al., 2008; Bond et al., 2013). Furthermore, such measurements rely on





converting absorption to BC mass, which can vary significantly and is a major strand of current research into BC (e.g. Bond et al., 2013).

Determination of $ER_{OC}$ and $EF_{OC}$ using the cTOF-AMS relies on converting OM to OC using an uncertain ratio, which typically ranges from 1.4 for fresh urban aerosol to 2.2 for aged non-urban aerosol (Turpin and Lim, 2001). We used a value of 1.6 in this study, which is considered typical for fresh biomass burning (Yokelson et al., 2009; Akagi et al., 2012). Given the differences between our reported $EF_{OC}$ and those in the literature are much larger than the range in OM/OC previously observed, this is unlikely to be a major driver of the differences reported here. Due to the significant concentrations of OA relative to inorganic species in this environment, the default fragmentation table (Allan et al., 2004) used to apportion measured signals in the cToF-AMS to gas and particle phase chemical components is modified based on calibrating the response of the instrument to sulphate. This methods follows established protocols for biomass burning (Ortega et al., 2013), yielding a change in OA of only a few percent and can therefore be discounted as a potential major source of bias and uncertainty in our reported $ER_{OC}$ and $EF_{OC}$.

The SP2 measures BC mass directly without relying on converting from absorption, so is likely better suited to sampling in this environment. The SP2 however will not detect BC containing particles with a diameter less than 60nm, which would bias the reported BC mass concentrations lower. The closed BC-mass size distributions in Fig. 4 though render this unlikely without a very large amount of BC-containing particles below the SP2 size cut-off, which would be required to substantially increase the BC mass concentration. Furthermore, while we do not have SMPS size distributions in the plumes, in the near-field at the regional scale we do not observe a highly enhanced ultrafine mode which would be expected if there was a large BC contribution at these sizes. Kondo et al. (2011) studied BC emissions from biomass burning in North America and Asia using the SP2 instrument. While the study is not directly comparable to ours due to the emissions being a few hours old rather than a few minutes old and from a different environment, they found BC emission ratios were similarly lower (factor of 2-5) than other literature values. This would support a potential reason for the differences between our reported BC emissions being at least in part due to differences in the measurement techniques. However, May et al. (2014) when studying prescribed fires in the United States found that laboratory and airborne derived EFs using a SP2 were generally higher than values previously reported in the literature. Uncertainties relating to the SP2 instrument response to different types of BC is a potential source of bias given that the instrument was calibrated using a reference standard for urban anthropogenic BC rather than one specific to biomass burning. Future studies in both the laboratory and field environments utilising a range of measurement techniques would be highly beneficial in terms of examining potential biases in different methods.

### 4.3 Implications

Global and regional numerical models are typically unable to reproduce aerosol optical depth (AOD) using standard configurations for emission inventories without scaling emissions by factors that vary both model-to-model and region-by-region (Kaiser et al., 2012; Tosca et al., 2013). Scaling factors can range from 1.5-5, representing a significant under-prediction of aerosol abundance in the atmospheric column. The EF values presented here are generally either similar to or lower than previous values reported for this biomass burning environment, suggesting that the EFs used in models are not responsible for





the underestimate of AOD over tropical South America; several modelling studies have been undertaken during SAMBBA (Archer-Nicholls et al., 2015; Reddington et al., 2016; Pereira et al., 2016; Johnson et al., 2016) and have required scaling of their emissions to match in-situ and satellite measurements. Consequently, scaling emission factors to match observations implies that the discrepancy lies elsewhere if it does relate to emissions (e.g. fire detections being biased low, uncertainties in

the evolution of fires), or other aspects of models such as their processes and assumptions.

We observe significant contrasts between the chemical components emitted by the Rondônia and Tocantins fires that are consistent with the differences in fuel and combustion efficiency of the fires. The Tocantins fire emitted 18% more $CO_2$ than the Rondônia fire, while for the particulate phase species, 97% of the total mass for the Rondônia rainforest fire was composed of organic aerosol compared to 84% for the Tocantins Cerrado fires. These results illustrate how the combustion efficiency and

10 fuel content of a fire can strongly influence the composition of the emissions, particularly in the case of the relative contribution of BC. Such contrasts will strongly control the single scattering albedo of the emitted smoke (e.g. Pokhrel et al., 2016) and perturb atmospheric heating rates and radiative forcing. Greater relative emissions of OA can significantly affect cloud droplet formation given that 45-75% of biomass burning OA has been shown to be water soluble (Reid et al., 2005; Asa-Awuku et al., 2008), which can again perturb the radiative balance of the atmosphere. Consequently, the initial conditions at source can

potentially play a large role in determining the weather, climate and air quality implications of the significant atmospheric burden of biomass burning across the region.

## 5   Conclusions

In-situ observations of near-field biomass burning emissions from two distinct fire types in Brazil are presented and evaluated. We presented fire integrated emission ratios and emission factors from a large smouldering rainforest wildfire in Rondônia state

and numerous smaller man-made flaming Cerrado fires in Tocantins state. The two fires differed substantially in emissions of CO and $CO_2$, resulting in MCEs of 0.79 and 0.94 for the Rondônia and Tocantins fires respectively. OA emissions also varied with the Rondônia smouldering rainforest fire having a higher emission factor for OA than the Tocantins flaming Cerrado fires, with OA comprising 97% of the emitted sub-micron mass in the former and 84% in the latter. The BC emission per kg fuel burnt was an order of magnitude higher for the Tocantins fires than the Rondônia fire. These results illustrate that the initial

fire conditions can play a significant role in determining the impacts on the Earth system by biomass burning emissions. In particular, the relative contribution of BC can vary significantly, which will represent a major control on the single scattering albedo of the aerosol burden over a given region and fire regime.

Compared with previous deforestation fire EFs in the literature and in emission inventories, the Rondônia particulate emissions differ substantially, with the only exception being the EF value for OA. This was likely due to the bias towards smoul-

30 dering emissions of the wildfire, which represents the lower extreme in terms of combustion efficiency compared to previous deforestation fire measurements. Gas phase EFs for the Cerrado environment suggest that the fires are representative of previous measurements in the literature. However, particulate emission factors for the Tocantins fire were 2-4 times lower for BC and OA than those reported in the literature for Cerrado or savannah type fires. One potential reason for this discrepancy





is the different measurement techniques used in this study, which measure OA and BC more directly than the filter-based measurements typically used in past studies. We recommend that comparisons of techniques are made in the future to assess the size of any such potential biases. Our calculated EFs do not indicate that the scaling of emissions that is required within global and regional numerical models to reproduce in-situ and satellite aerosol concentrations over Brazil (Kaiser et al., 2012;

Tosca et al., 2013; Archer-Nicholls et al., 2015; Reddington et al., 2016; Pereira et al., 2016; Johnson et al., 2016) is related to underestimates in EFs used in emission inventories.

**Data availability**

All raw time series data used to derive the emission ratios and factors from the FAAM research aircraft are publically available from the Centre for Environmental Data Analysis website (http://www.ceda.ac.uk/). SP2 size distribution data is available

on request due to the size of the data files. Data masks for categorising flight patterns into plume-sampling and other sampling types (vertical profiles and SLRs) are currently available on request. Active fire data used in the manuscript is available publically from NASA (see acknowledgements for further details).

*Author contributions.*  A. K. Hodgson and W. T. Morgan analysed the data and wrote the manuscript. S. O'Shea, J. D. Allan, E. Darbyshire, D. Liu and J. Lee provided additional data analysis support, including data processing and quality assurance. S. Bauguitte and M. J. Flynn

operated the gas-phase and aerosol instruments respectively during the field campaign. B. Johnson, J. Haywood, K. M. Longo, P. E. Artaxo and H. Coe led the planning of the field campaign and were co-principal investigators on the SAMBBA project.

*Acknowledgements.*  We would like to acknowledge the substantial efforts of the whole SAMBBA team before, during and after the project. Airborne data was obtained using the BAe-146-301 Atmospheric Research Aircraft (ARA) flown by Directflight Ltd and managed by the Facility for Airborne Atmospheric Measurements (FAAM), which is a joint entity of the Natural Environment Research Council (NERC)

and the Met Office. Active fire data was produced by the University of Maryland and acquired from the online Fire Information for Resource Management System (FIRMS; https://earthdata.nasa.gov/data/near-real-time-data/firms/abouts; specific product: MCD14ML). E. Darbyshire was supported by NERC studentship NE/J500057/1 and NE/K500859/1. This work was supported by the NERC SAMBBA project NE/J010073/1.



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







**Figure 1.** Overview of case studies used in the analysis with the Rondônia flight (B737) shown on the left and the Tocantins flight (B742) shown on the right. Panel a. Photographs taken from the aircraft of the Rondônia and Tocantins fires courtesy of William T. Morgan and Axel Wellpott respectively. Panel b. Low-level flight tracks including MODIS hotspot data from TERRA and AQUA overpasses for the dates of interest. Plume interceptions are also marked. 10 km$^2$ box represents scale for flight-track. Panel c. Time series of CO, rBC and OA during the near-field fire sampling periods of the flight.





**Figure 2.** Relationship between excess concentrations of trace gas and particulate phase species relative to excess carbon monoxide for the Rondônia and Tocantins fires. Solid lines show line of best fit from linear regressions forced through zero.



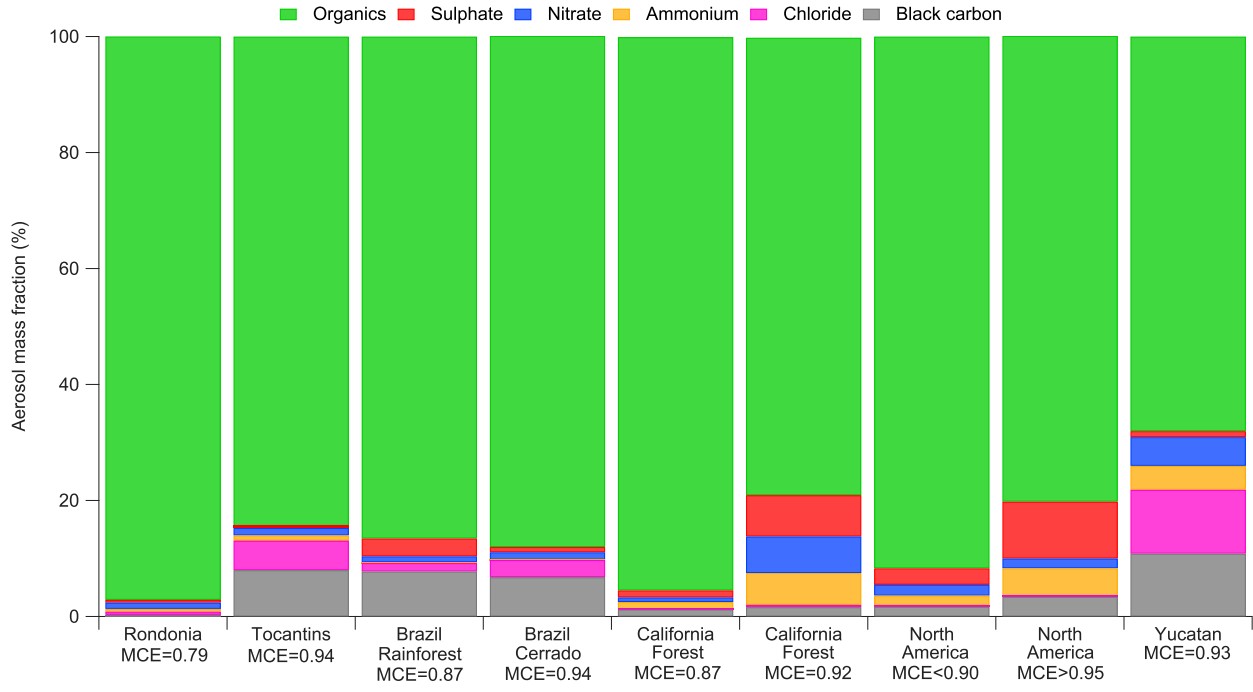

**Figure 3.** Mass fraction of aerosol components as a percentage of total aerosol mass, including black carbon, organic aerosol, chloride, ammonium, sulphate and nitrate for the Rondônia fire, Tocantins fires and values for previous studies from the Yucatan (Yokelson et al., 2003), Brazil (Ferek et al., 1998), California (Sahu et al., 2012) and North America (Kondo et al., 2011).





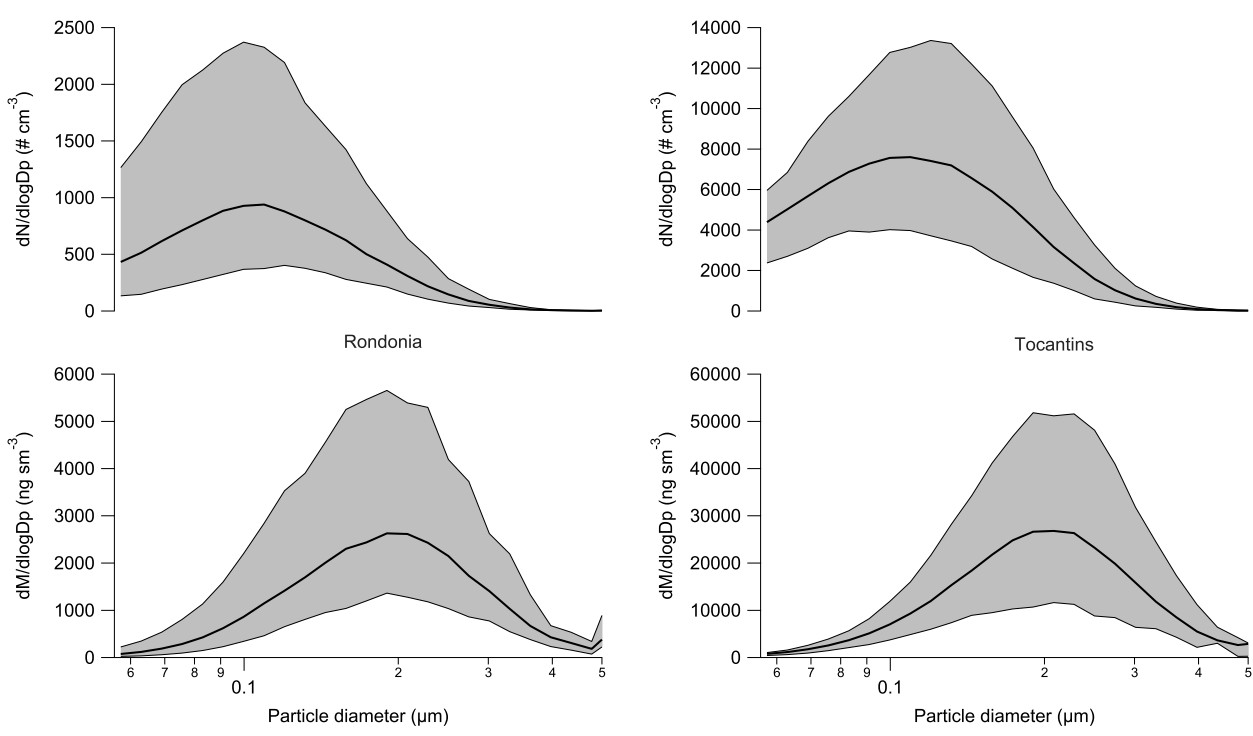

**Figure 4.** Black carbon number and mass-size distributions for in plume measurements of the Rondônia and Tocantins fires. The grey bound shading indicates the minimum and maximum size distributions from the plume intercepts on each flight and the solid black line gives the average over all plume interceptions. Note the difference in scales between Rondônia and Tocantins size distributions.

**Table 1.** Modified Combustion Efficiency (MCE) for the Rondônia fire and the Tocantins fires. Calculation methods as described in section 2.2.2. Also included are other studies MCE's from Brazil and from other locations with the specific fuel type quoted. [1]This study, [2]Yokelson et al. (2007), [3]Christian et al. (2007), [4]Ferek et al. (1998), [5]Kaufman et al. (1992), [6]Yokelson et al. (2009), [7]Yokelson et al. (2003), [8]Bertschi et al. (2003), [9]Akagi et al. (2011), [10]McMeeking et al. (2009).

| | Modified Combustion Efficiency | | |
| --- | --- | --- | --- |
| Study | Rainforest-like | Cerrado-like | Mixed |
| **Rondônia: Rainforest[1]** | **0.79 ± 0.02** | | |
| **Tocantins: Cerrado[1]** | | **0.94 ± 0.02** | |
| Brazil: forest, pasture, grass average[2] | | | 0.91 ± 0.02 |
| Brazil: Residual smouldering combustion of logs[3] | 0.799 ± 0.059 | | |
| Brazil: Forest residual smouldering and Cerrado[4] | 0.87 | 0.94 | |
| Brazil: Deforestation and Cerrado[5] | 0.91 | 0.97 | |
| Yucatan: Deforestation and Crop residue[6] | 0.927 ± 0.013 | 0.934 ± 0.023 | |
| Africa: Savannah[7] | | 0.938 ± 0.019 | |
| Laboratory study: smouldering cotton wood and Zambian logs[8] | 0.854 | | |
| California: Chaparral[9] | | 0.933 | |
| Laboratory study: range of fuel types[10] | | | 0.857-0.977 |





**Table 2.** Emission ratios (ppmv/ppmv) and uncertainties for CO and $CH_4$ with respect to CO ($ER_{x/CO}$) and $CO_2$ ($ER_{x/CO_2}$) for the Rondônia fire and the Tocantins fires. Uncertainties for the two fires shown are one standard deviation in the line of best fit. Calculation methods are described in section 2.2.3. Also included are other ERs from other studies conducted in Brazil and other locations with the specific fuel type quoted. [1]This study, [2]Christian et al. (2007), [3]Yokelson et al. (2009), [4]Bertschi et al. (2003), [5]Yokelson et al. (2003), [6]Akagi et al. (2012), [7]Wooster et al. (2011), [8]Crutzen et al. (1985).

| Study | $ER_{CO/CO_2}$ (x100) | $ER_{CH_4/CO_2}$ (x1000) | $ER_{CH_4/CO}$ (x1000) |
|---|---|---|---|
| *Rainforest-like* | | | |
| **Rondônia: Rainforest**[1] | **25.8 ± 0.70** | **9.8 ± 0.20** | **38.0 ± 0.80** |
| Brazil: Residual smouldering combustion of logs[2] | 27.5 ± 9.3 | - | 14.3 ± 8.60 |
| Yucatan: Deforestation and Crop residue[3] | - | - | 110.4 |
| Laboratory study: smouldering cotton wood and Zambian logs[4] | 16.85 | - | 219 |
| *Cerrado-like* | | | |
| **Tocantins: Cerrado**[1] | **6.8 ± 0.30** | **3.6 ± 0.03** | **53.1 ± 1.60** |
| Africa: Savannah[5] | 6.64 ± 2.14 | - | 53.1 ± 11.8 |
| California: Chaparral[6] | 7.13 ± 0.55 | - | 87.2 ± 2.4 |
| Africa: Savannah[7] | 9.60 ± 3.10 | 4.30 ± 1.70 | 4.60 ± 0.70 |
| *Mixed* | | | |
| Brazil: Mix of fire types[8] | 15.4 | 1.2 | - |



**Table 3.** Trace gas emission factors ($g\,kg^{-1}$ of dry fuel burned) for Rondônia and Tocantins fires determined using the calculations as shown in section 2.2.4. EF values from previous studies in Brazil are included for comparison with the specific fuel type stated. Also included are EFs from different geographical locations around the world and values used in GFEDv3/GFASv1.0 and GFEDv4 emission inventories (van der Werf et al., 2010; Kaiser et al., 2012). [1]This study, [2]Christian et al. (2007), [3]Ferek et al. (1998), [4]Ward et al. (1992), [5]Kaufman et al. (1992), [6]Yokelson et al. (2009), [7]Akagi et al. (2011), [8]Andreae and Merlet (2001), [9]Bertschi et al. (2003), [10]GFEDv3, [11]GFEDv4, [12]Sinha et al. (2003), [13]Yokelson et al. (2003), [14]Wooster et al. (2011), [15]McMeeking et al. (2009).

| Study | $CO_2$ | CO | $CH_4$ |
|---|---|---|---|
| *Rainforest-like* | | | |
| **Rondônia: Rainforest[1]** | **1447 ± 148** | **237 ± 24** | **5.17 ± 0.53** |
| Brazil: Residual smouldering combustion of logs[2] | 1346 ± 123 | 229 ± 64.6 | 17.1 ± 10.0 |
| Brazil: Forest smouldering[3] | 831 ± 22 | 120 ± 13 | 12.5 ± 6.1 |
| Brazil: Deforestation[4] | 1614 | 110 | 6.6 |
| Brazil: Deforestation[5] | 1664 | 89 | 5.0 |
| Yucatan: Deforestation and Crop residue[6] | 1641 ± 40 | 80.18 ± 19.4 | 5.059 ± 2.369 |
| Global: Tropical forest[7] | 1643 ± 58 | 93 ± 27 | 5.07 ± 1.98 |
| Global: Tropical Forest[8] | 1580 ± 90 | 104 ± 20 | 6.8 ± 2.0 |
| Laboratory study: smouldering cotton wood and Zambian logs[9] | 1461.5 | 156.5 | 19.7 |
| GFEDv3/GFASv1.0: Deforestation[10] | 1626 | 101 | 6.6 |
| GFEDv4: Deforestation[11] | 1643 | 93 | 5.07 |
| *Cerrado-like* | | | |
| **Tocantins: Cerrado[1]** | **1711 ± 175** | **74 ± 8** | **2.23 ± 0.23** |
| Brazil: Cerrado[3] | 928 ± 30 | 57 ± 28 | 3.7 ± 2.7 |
| Brazil: Cerrado[4] | 1722 | 58 | 1.3 |
| Brazil: Cerrado[5] | 1783 | 24 | 0.6 |
| Global: Savannah[7] | 1686 ± 38 | 63 ± 17 | 1.94 ± 0.85 |
| Global: Savannah and Grassland[8] | 1613 ± 95 | 65 ± 20 | 2.3 ± 0.9 |
| Africa: Savannah[12] | 1700 ± 60 | 68 ± 30 | 1.7 ± 0.98 |
| Africa: Savannah[13] | 1703 ± 39 | 71.5 ± 21.7 | 2.19 ± 1.0 |
| African: Savannah[14] | 1665 ± 54 | 101 ± 30 | 2.5 ± 0.9 |
| GFEDv3/GFASv1.0: Savannah[10] | 1646 | 61 | 2.2 |
| GFEDv4: Savannah[11] | 1686 | 63 | 1.94 |
| *Mixed* | | | |
| Laboratory study: range of fuel types[15] | 1034 ± 175 to 1868 ± 5 | 43.0 ± 1.4 to 129.5 ± 4.9 | 0.2 to 5.9 ± 1.2 |





**Table 4.** Mass fraction of aerosol components as a percentage of total aerosol mass, including black carbon, organic aerosol, chloride, ammonium, sulphate and nitrate for the Rondônia fire, Tocantins fires and values from previous studies. Ferek et al. (1998) and Yokelson et al. (2009) reported a percentage total including other particulate matter that we did not measure in our study, therefore we recalculated the values presented just including BC, organics and inorganics in order to compare values.*OA for the two Brazilian values estimated assuming OM/OC is 1.6 in fresh smoke (Yokelson et al., 2009; Akagi et al., 2012). [1]This study, [2]Ferek et al. (1998), [3]Sahu et al. (2012), [4]Kondo et al. (2011), [5]Yokelson et al. (2009).

| Study | OM | BC | $SO_4^{2-}$ | $NO_3^-$ | $NH_4^+$ | $Cl^-$ |
|---|---|---|---|---|---|---|
| *Smouldering fires* | | | | | | |
| **Rondônia**[1] | **97.1** | **0.30** | **0.49** | **1.08** | **0.49** | **0.53** |
| Brazil: Forest[2] | 86.6 | 7.75 | 3.07 | 1.02 | 0.15 | 1.46 |
| California: Forest[3] | 95.4 | 1.2 | 1.2 | 0.8 | 1.2 | 0.10 |
| North America: Boreal forest[4] | 91.7 | 1.7 | 2.8 | 1.9 | 1.7 | 0.2 |
| *Flaming fires* | | | | | | |
| **Tocantins**[1] | **84.4** | **7.99** | **0.34** | **1.29** | **0.92** | **5.08** |
| Brazil: Cerrado[2] | 88.12 | 6.76 | 0.9 | 1.29 | 0.05 | 2.98 |
| California: Forest[3] | 78.9 | 1.6 | 7.1 | 6.3 | 5.5 | 0.4 |
| North America: Boreal forest[4] | 80.3 | 3.4 | 9.8 | 1.7 | 4.8 | 0.1 |
| Yucatan: Deforestation & Crop residue[5] | 68.0 | 10.8 | 1.02 | 4.95 | 4.14 | 11.01 |





**Table 5.** Emission ratios and uncertainties for particulate species with respect to CO ($ER_{x/CO}$) and $CO_2$ ($ER_{x/CO_2}$) for the Rondônia fire and the Tocantins fires. The ERs are presented as molar ratios and are multiplied by 1000. Calculation methods are described in section 2.2.3. Also included are other ERs from other studies conducted in Brazil and other locations with the specific fuel type quoted.[1]This study, [2]Akagi et al. (2012): California Chapparral, [3]Yokelson et al. (2009): Mix of crop residue and deforestation, [4]Jolleys et al. (2012), [5]Capes et al. (2008) using OM/OC ratio of 1.4.

| Study | ER | OM | OC | BC | $Cl^-$ | $NO_3^-$ | $SO_4^{2-}$ | $NH_4^-$ |
|---|---|---|---|---|---|---|---|---|
| **Rondônia**[1] | $x/CO_2$ | **20.3** | **12.7** | **0.05** | **0.10** | **0.20** | **0.09** | **0.08** |
| | | **±6.1** | **±3.8** | **±0.02** | **±0.03** | **±0.06** | **±0.03** | **±0.02** |
| **Tocantins**[1] | | **2.8** | **1.7** | **0.28** | **0.19** | **0.04** | **0.01** | **0.03** |
| | | **±0.8** | **±0.5** | **±0.08** | **±0.06** | **±0.001** | **±0.003** | **±0.009** |
| California[2] | | 3.55 | 2.22 | 0.783Z | 0.0497Z | 0.0961 | 0.00358 | 0.06 |
| | | ±0.857 | ±0.536 | ±0.536 | ±0.536 | ±0.536 | ±0.00328 | ±0.0395 |
| **Rondônia**[1] | $x/CO$ | **123.4** | **77.0** | **0.30** | **0.67** | **1.21** | **0.52** | **0.5** |
| | | **±37.1** | **±37.1** | **±0.09** | **±0.2** | **±0.36** | **±0.16** | **±0.2** |
| **Tocantins**[1] | | **68.2** | **42.6** | **6.1** | **4.62** | **1.03** | **0.28** | **0.79** |
| | | **±20.5** | **±12.8** | **±1.8** | **±1.39** | **±0.31** | **±0.08** | **±0.24** |
| Yucatan[2] | | - | 26.4 | 6.3 | 6.3 | 2.9 | 0.6 | 2.4 |
| Northern Australia[4] | | 329±23.0 | - | - | - | - | - | - |
| SE Mexico City[4] | | 51.0±1.0 | | | | | | |
| West Africa[5] | | 65.0±2.0 | 41.0±2.0 | 7.2±0.9 | - | - | - | - |





**Table 6.** Particulate species emission factors (g kg$^{-1}$ of dry fuel burned) for Rondônia and Tocantins fires determined using the calculations as shown in section 2.2.4. EF values from previous studies in Brazil are included for comparison with the specific fuel type stated. Also included are EF's from different geographical locations around the world and values used in GFEDv3/GFASv1.0 and GFEDv4 emission inventories (van der Werf et al., 2010; Kaiser et al., 2012). [1]This study, [2]Ferek et al. (1998), [3]Yokelson et al. (2009), [4]Akagi et al. (2011), [5]Andreae and Merlet (2001), [6]van der Werf et al. (2010); Kaiser et al. (2012), [7]Sinha et al. (2003), [8]McMeeking et al. (2009).

| Study | OM | OC | BC | Cl$^-$ | NO$_3^-$ | SO$_4^{2-}$ | NH$_4^-$ |
|---|---|---|---|---|---|---|---|
| *Rainforest-like* | | | | | | | |
| **Rondônia**[1] | **8.00** | **5.00** | **0.019** | **0.04** | **0.078** | **0.034** | **0.033** |
| | **±2.53** | **±1.58** | **±0.006** | **±0.01** | **±0.025** | **±0.011** | **±0.011** |
| Brazil: Forest smouldering[2] | - | 17.9 | 1.5 | - | - | - | - |
| | | ±7.6 | ±0.9 | | | | |
| Yucatan: Deforestation & Crop residue[3] | 3.254 | 2.117 | 0.541 | 0.509 | 0.233 | 0.047 | 0.192 |
| | ±0.690 | ±0.569 | ±0.569 | ±0.377 | ±0.056 | ±0.024 | ±0.136 |
| Global: Tropical forest[4] | - | 4.71 | 0.52 | 0.15 | 0.11 | 0.13 | 0.00564 |
| | | ±2.73 | ±0.28 | ±0.16 | ±0.05 | ±0.088 | ±0.0172 |
| Global: Tropical Forest[5] | - | 5.2 | 0.66 | - | - | - | - |
| | | ±1.5 | ±0.31 | | | | |
| GFEDv3/GFASv1.0: Deforestation[6] | - | 4.3 | 0.57 | - | - | - | - |
| GFEDv4: Deforestation[6] | - | 4.71 | 0.52 | - | - | - | - |
| *Cerrado-like* | | | | | | | |
| **Tocantins**[1] | **1.31** | **0.82** | **0.13** | **0.09** | **0.013** | **0.0006** | **0.015** |
| | **±0.42** | **±0.26** | **±0.04** | **±0.03** | **±0.004** | **±0.0002** | **±0.005** |
| Brazil: Cerrado[2] | - | 5.9 | 0.7 | - | - | - | - |
| | | ±2.8 | ±0.4 | | | | |
| Africa: Savannah[7] | - | 2.3 | 0.39 | 0.97 | 0.16 | 0.17 | - |
| | | ±1.2 | ±0.19 | ±1.4 | ±0.11 | ±0.18 | |
| Global: Savannah[4] | - | 2.62 | 0.37 | 0.23 | 0.016 | 0.018 | 0.0035 |
| | | ±1.24 | ±0.20 | ±0.055 | ±0.013 | ±0.009 | ±0.0035 |
| Global: Savannah & Grassland[5] | - | 3.4 | 0.48 | - | - | - | - |
| | | ±1.4 | ±0.18 | | | | |
| GFEDv3/GFASv1.0: Savannah[6] | - | 3.2 | 0.46 | - | - | - | - |
| GFEDv4: Savannah[6] | - | 2.62 | 0.37 | - | - | - | - |
| *Mixed* | | | | | | | |
| | | 0.5 | - | 0.03±0.02 | 0.0±0.0 | 0.01 | 0.0±0.01 |
| Laboratory study: range of fuel types[8] | | to | | to | to | to | to |
| | | 44.2 | - | 5.39 | 0.84±1.17 | 0.73±0.34 | 0.51 |