# Peer review of "Near-field emission profiling of Tropical Forest and Cerrado fires in Brazil during SAMBBA 2012"

_Atmospheric Chemistry and Physics, 2016_

## Referee Comment (RC1) · R. J. Yokelson (Referee) · 6 Feb 2017

Review of acp-2016-1019

By Bob Yokelson

Several savanna fires and one forest fire were sampled in the near-field with a heavily instrumented aircraft in Brazil to measure emission factors (EF). These are the first EF measurements in Brazil since 2004 and 1996. The earlier studies sampled more fires, but there is way too little sampling of these extremely important, but variable sources. Further, this study used some instruments not previously deployed on fires in Brazil. The data should be published without a doubt. Since fires are variable, any EFs measured will fall on a Bell curve. Thus, the expectation is not that this data must agree with previous averages, but that it can be combined with previous data to

nudge an evolving global average. At the same time, the authors find no evidence that erroneously low EF in previous work caused the ubiquitous low a-priori emissions in regional to global models.

There is one serious oversight that needs to be fixed. Ferek et al., (1998) reported EF as gC/KgC not g of species /Kg fuel. Assuming the fuel is about 50% C, their EF for BC need to be divided by two to compare to the work here. The EF for $CO_2$ should be divided by two and then multiplied by (44/12) to convert to g $CO_2$/ kg fuel, etc. The authors should recalculate all the Ferek et al numbers correctly and then update their comparisons, which come much closer in many cases if this step is taken.

Minor terminology point: "rainforest" is kind of colloquial. I suggest evergreen tropical forest throughout as distinct from seasonally dry tropical forest or just tropical forest if they are not sure which type. It is important to distinguish between understory and deforestation fires, which they do.

One topic not mentioned that might make a good addition to the paper if possible. Were any aerosol optical properties measured? If so, did they scale with EC/OC or MCE as in Pokhrel et al., (2016)?

Some specific comments in order of appearance or occurrence to reviewer.

P1, L12: I suggest defining organic aerosol as "(OA)". OA is the primary measurement, but OC is reported by dividing OA by 1.6. Reporting OC facilitates comparison with more historical data, but it might be worth reporting OA too?

P1, L15: "fuel content" to "fuel mixture" maybe?

P1, L17: Perhaps change "scaling" to "scaling up" to set the context for why the possibility of low EF was interesting.

P1, L18: Maybe simplify the end of this sentence as e.g. ". . . . one potential cause of low a-priori emissions in modeling studies."

P2, L12: delete "to"

P2, L15: There is a far more complex mix of burning in Brazil than just two fire types as discussed at length elsewhere (section 2.3, Yokelson et al., 2007). Referees complained about the length of that section, but maybe a broader summary is in order here. Something like ~"a range of climate and fire types occurs in Brazil and fire-impacted ecosystems include pure grassland, a gradient of wooded savannah into dry (seasonal) tropical forest (aka Cerrado), and evergreen tropical forest (Ward et al., 1992). In forested areas understory fires can occur, but deforestation fires to establish pastures or croplands, along with pasture maintenance and agricultural residue fires are the most common types of burning on an evolving heterogeneous landscape." You could cite our 2007 paper or any of the excellent papers cited there-in. Then mention (move here) the later statements (already with citations) about burning shifting from forests to savannas.

P2, L23: Actual landscape fires can rarely be characterized by a single stage, but instead there is a dynamic, variable mix of flaming and smoldering processes.

P2, L29-30: By sampling more we get a better idea of the mean and range. We can also learn about the factors driving the variability, which I think this study may do thru its analysis.

P3, L1: Insert "and other" before "regions" as the mix of AMS and SP2 been used in the US on chaparral fires (Akagi et al., 2012), prescribed forest fires (May et al., 2014), agricultural fires (Liu et al 2016), etc.

P3, L1-4: The historical context and the contrast with the current effort is a bit oversimplified. Previous work was not necessarily less sensitive or unable to probe individual plumes. Yokelson et al., (2007) flew Artaxo's calibrated nephelometer to get real-time PM10 in numerous single plumes and computed PM10 EFs for Brazilian fires. Light scattering measurements are pretty sensitive. It could be that PM based on light scattering is not as accurate as an AMS, but evidently a nephelometer was used to scale

the AMS data in this work. In SCAR-B in Brazil and SAFARI 2000 in Africa, a very large bag sampler was used and instantaneously filled in numerous individual plumes at multiple ages per plume. This allowed sensitive filter sampling, though filter artifacts can occur (e.g. Ferek et al., 1998; Sinha et al., 2003). Also, real-time data was acquired with a suite of PM instruments in individual plumes in the Mexican tropics in Yokelson et al., (2007, 2009, 2011). I think Capes et al., (2008) deployed an AMS on North African fires. To my knowledge this was the first study to use both AMS and SP2 in individual smoke plumes in the tropics. However, the sentence implying this is particularly true for airborne work should be deleted unless the authors can cite a ground-based study of individual tropical BB plumes that used both SP2 and AMS. Finally, depending on performance, different approaches have a different set of strengths and weaknesses.

P4, L1-5: The authors estimated the mass of PM from scattering data to scale the AMS mass. They should report what mass scattering efficiency they used and an error estimate. Then a scaling factor of 2.69 +/- ? Further, if the SMPS could be used to measure the AMS collection efficiency, it seems like it could also be used for an independent estimate of the AMS scaling factor. On L5 add "forest fire" before "data".

P4, L15: It's not super clear how coincidence was solved. Evidently they did not account for small rBC particles below the SP2 size cutoff as is sometimes done, but the figures seem to indicate this correction is not needed in this case.

P5, L15-20: Perhaps report both OA and OC and use OA along with the other species to estimate and compare fine PM?

P5, L23: Technically "hydrocarbons" should be non-methane organic gases (NMOGs) since O-containing VOC dominate. Likewise on P6, L17

P6, L2-3 don't need, smoke dilution rates slow down exponentially with time and can be even faster on ground-based measurements closer to the source.

P6, L22: A "few" percent is probably more accurate than 1-2% if particulate carbon,

which was measured is not included in total C.

P6, L26 "molecular" should be "atomic in the case of carbon (an error I have made many times).

P7, L15: This is a nice description of the fire. Hotspots are shown in the figure on the day of sampling. It is stated there were no hotspots the previous day and the fire may have started the same day it was sampled. The fire seems a bit large to emerge in one morning and the authors could check a few of the previous days for hotspots since, due to cloud cover or orbital gaps any particular day may have no hotspot data. Brazilian fires can be anthropogenic and off-road due to the presence of indigenous peoples. We likely sampled some of these in an indigenous preserve along the Xingu River.

L26: 30 ppm is really fresh! That's a good sign that the EF are not distorted by mixing.

P8, L24: The high r-squared likely indicates that the plume was well-mixed and the fire burned a homogeneous fuel bed as opposed to a "common source." (relevant also to P10, L8)

P8, L25-26: Perhaps the authors are trying to say that there is more variability when sampling a group of fires than a single fire, but the group of fires has a fairly high r-squared nonetheless.

P9. L4-5: Also a general comment. The data for the Rondonia fire is in good agreement for PM and $CH_4$ with the data for tropical dry forest fires in Mexico (Table 3 in Yokelson et al., (2011)).

Yokelson, R. J., Burling, I. R., Urbanski, S. P., Atlas, E. L., Adachi, K., Buseck, P. R., Wiedinmyer, C., Akagi, S. K., Toohey, D. W., and Wold, C. E.: Trace gas and particle emissions from open biomass burning in Mexico, Atmos. Chem. Phys., 11, 6787-6808, doi:10.5194/acp-11-6787-2011, 2011.

P10, L2: It's worth clarifying if they are showing the size of the BC core or the core plus coating, and also confirming that coated BC particles were included in the computation

of BC mass.

P10, L15: I would delete "more than" and possibly add "approximately" in light of the uncertainties.

P10, L25 – P11, L6: Several things can be quickly fixed in here and throughout. The Ferek et al values that are compared to need to be converted to g species per kg of fuel. After doing that, the Akagi et al review paper lists these values for Ferek et al as examples: Savanna (OC 2.94, BC 0.35 (note good agreement with Sinha et al., (2003) for African savannas)) and Tropical Forest (OC 4.34, BC 0.46). So for instance, on P11, L3, the Ferek et al value is 0.35 and not 0.7 for EFBC. The authors should go thru all the text, tables, and also figures if applicable, and update the values quoted for Ferek et al and the comparisons as well. Further, most of the "BC" measurements they compare to are actually "EC" measurements that can be impacted by charring of OC. (The PAX measurement in Stockwell et al are an exception.) This should probably be mentioned here (along with later mention) and might inform the comparison.

P11, L2: The Stockwell measurements were by in-situ photoacoustic spectroscopy. This may be a good place to mention EC artifacts and the possibility of lower EFBC because of an unusually smoldery fire. Nonetheless, the data can be factored into the global average nudging it down.

P12, L22: change "to" to "as"

P13, L9: change "is" to "was"

P13, L26: Re "the instrument" – was the SAMBBA SP2 or the May et al SP2 or both SP2's calibrated with urban-BC relevant material? Suggest clarifying by changing to "our instrument" "both instruments", etc as appropriate.

P13, L27-28: Informational only, we have just completed such a comparison in FIREX

P14, L19: If there is no clearing then a fire is not a deforestation fire, but understory fires can be indigenous/anthropogenic to promote favored tree species, for hunting, or

to improve access.

P14, L20: "numerous" may be a bit overstated for what looks like about ten hotspots in 3 groups. Maybe "several" or "a group of" is better.

P14, L24: "illustrate" to "confirm" since the differences in initial emissions is a fairly well-known topic.

Fig. 1. Nice fire pics, perhaps a larger version in supplement would be worthwhile?

Fig. 4. It seems odd that the values are larger for the Tocantins fires given the lower emissions??

Table 1: header, reference 9 is Akagi et al., (2012). "laboratory" entry could be "lab/field" since the cottonwood log was in lab, but the Zambian log was in the field.

Table 2: There should not be any missing ("-") values since all three gases were measured in all cited studies. For the Brazil smoldering logs (ref 2) the value for $CH4/CO$ (X1000) is 143 not 14.3. This value for ref 7 may also be a factor of ten low?

Table 3: Third and thirteenth entries down for $EFCO_2$ (ref 3) looks suspicious.

---

## Referee Comment (RC2) · G R McMeeking (Referee) · 13 Feb 2017

The manuscript presents biomass burning emission measurements during a series of research flights over several fires in Brazil. This is a region of global significance in terms of both total particulate matter as well as black carbon emissions. While there have been several previous campaigns focused on characterizing emissions in this region, none have used more modern instruments, and the relative scarcity of data from this area coupled with its importance certainly merits publication in my opinion.

I have little to add beyond Dr. Yokelson's thorough comments, but do recommend addressing a couple of smaller issues/areas in more detail in the manuscript in a revised version:

It would be helpful to others in the SP2 community to know a bit more about how the

instrument was operated in these more challenging conditions. Some small details regarding the sample flow rate and dilution (if any) could be provided. In addition, an estimate of the concentration limit where true particle coincidence (multiple BC present in the laser beam at the same time) would be helpful, and a verification that the field data remained below this value.

The manuscript mentions the use of dryers on the AMS and SP2 inlet line, but does not discuss losses. Deriving correction factors for losses in nafion dryers can be difficult, but some short discussion of potential impacts on measurement uncertainties would be useful. On a related point, I assume the nephelometer inlet line RH was at times quite different from the AMS/SP2 line, else the nephelometer could be used to apply corrections for the pin-hole issue earlier in the study at all times. I am curious if the pinhole blockage effects may have varied with sample line RH. Any systematic relationships between emission ratios to CO and RH might hint at this.

Suggest referring to BC "core" diameters rather than BC diameter to avoid potential confusion with the mixed particle size.

I recommend reporting an average OA mass concentration in Table 6. This can be useful for any future comparisons with EF measured at different concentrations and can help untangle potential impacts of semi-volatile partitioning.

---

## Referee Comment (RC3) · C. Paton-Walsh (Referee) · 18 Feb 2017

This is an excellent paper presenting some very valuable measurements of emissions from fires in Brazil, a poorly sampled region of the globe. The paper is definitely suitable for publication in ACP, and has been expertly reviewed already by Bob Yokelson and Gavin McMeeking. I have a few minor additional comments below:

1. It has become traditional (following Yokelson et al.,[1999]), when calculating emission ratios via the best straight-line fit to a plot of one species against the reference species, to first subtract the background amounts and then force the regression to go through zero. However, subtracting background amounts is not required, because this has no mathematical impact on the gradient of the best line fit. Forcing the line through zero may change the gradient, but it puts unnecessary weight to the background concentrations measured/assumed. If these are very close to the real (and unchanging) background amounts, then the change to the gradient that occurs when you force the line through zero will be small. If the background assumed is incorrect, or is changing, the effect can be quite significant, as pointed out in a later paper by Yokelson et al., [2013]. A generalised least squares regression (that takes into consideration the uncertainties in both x and y) is a mathematically simpler and more accurate way to determine the emission ratio. I recommend this way to calculate emission ratios. It will not avoid all of the issues pointed out in Yokelson et al., [2013] if the background amounts are hugely variable, but it will minimise them compared to the calculation the authors have used in this study. Having said that, the high r-squared values lend confidence to the results in this study. If the authors are confident that they haven't biased their results significantly and do not wish to go back and recalculate the emission ratios, then I recommend that a sentence is added on this matter. The sentence should point out that forcing the regression through zero can bias the emission ratio if the background amounts assumed are wrong or change, but in this case they are confident they are not subject to the pitfalls described in Yokelson et al., [2013].

2. The use of the 1 sigma uncertainty of the best line fit as the total uncertainty in the emission ratio is not valid when the uncertainties in the individual points are correlated with one another (which they are in this case). Ideally you should undertake a proper uncertainty analysis of your measurements. As a minimum you should acknowledge that the uncertainties in each point are correlated and so your value of the uncertainties will be an underestimate (since it will include the random errors only).

3. Finally, I assume that the correction to the AMS data that was required as a result of the partial blockage of the inlet would have added to the measurement uncertainties? Again, if it is not feasible to undertake a proper uncertainty analysis, you should at least acknowledge this has not been done and mention the additional uncertainty in the text.

References Yokelson, R. J., M. O. Andreae, and S. K. Akagi (2013), Pitfalls with the use of enhancement ratios or normalized excess mixing ratios measured in plumes

to characterize pollution sources and aging, Atmos. Meas. Tech., 6(8), 2155-2158, doi:10.5194/amt-6-2155-2013.

Yokelson, R. J., J. G. Goode, D. E. Ward, R. A. Susott, R. E. Babbitt, D. D. Wade, I. Bertschi, D. W. T. Griffith, and W. M. Hao (1999), Emissions of formaldehyde, acetic acid, methanol, and other trace gases from biomass fires in North Carolina measured by airborne Fourier transform infrared spectroscopy, J. Geophys. Res.-Atmos., 104(D23), 30109-30125, doi:10.1029/1999jd900817.

---

## Referee Comment (RC4) · C. Paton-Walsh (Referee) · 20 Feb 2017

This comment is prepared jointly by Clare Paton-Walsh and Bob Yokelson and addresses the first comment made in the original review by Clare Paton-Walsh, recommending a generalised least squares regression (that takes into consideration the uncertainties in both x and y) as a more accurate mathematical method for determining the emission ratios than the method actually used in the study. In particular the use of forcing the intercept through zero (after subtracting background values) was questioned.

This comment initiated a detailed discussion of the analysis of aircraft data like those in the study by Hodgson et al., between us both (Bob Yokelson and Clare Paton-Walsh). At the end of these discussions, we have come to agreement that in fact the real overall uncertainty in fire emission factors due to the small fraction of global fires sampled, weaknesses of any sampling procedure, lack of data on the exact fuel that burned, un-measured species in the carbon mass balance, etc., is much larger than the relatively insignificant differences that occur due to the method used in this study (and similar ones) versus a full generalised least squares regression weighted by the individual uncertainties of each point in the regression.

Some aspects of these measurements that make this conclusion particularly valid for aircraft-based source characterization studies are:

1. There are typically a vastly greater number of measurements made in background conditions, than through the smoke plume. This means that, whilst the background values are not known with zero uncertainty, the background is very well-characterized compared to the other points in the regression. This makes forcing the zero intercept a reasonable approach.

2. Similarly, the uncertainty in the measurements of CO and CO2, (the reference species plotted on the x-axis) is usually very small compared to the uncertainty for the species plotted on the y axis. This means that a simple linear regression (that only minimises the sum of the square of the deviations in the y-axis), will yield very similar results to a full regression that minimises the sum of the square of the deviations in both x and y weighted by the uncertainty in each for each individual point.

3. Indeed in Yokelson et al., (1999) it was found that close to fires at high concentrations several approaches to retrieve emission ratios (ER) usually gave similar answers and within the uncertainties of each approach. Those authors chose to force the intercept in their data pair plots partly because of the following scenario that occurs on occasion. One can fly through the plume at low concentrations of "x" and measure a high "y/x" ratio (e.g, smoldering) and fly through the plume at high concentrations of "x" and get low "y/x" ratio (e.g. flaming). Using regression in this case without forcing the intercept gives a negative gradient. Using regression with the intercept forced still

gives a reasonable estimate of the ER.

Given these aspects of the aircraft data, the analysis undertaken should yield a reasonable result. We suggest that the authors may choose to ignore the first comment in the review by Clare Paton-Walsh when preparing their final manuscript.

Finally, it's worth briefly clarifying the main point of Yokelson et al., (2013). Changing the dilution air composition as mixing proceeds can have very large effects on the "apparent ER" that would be measured downwind. This is especially problematic when changes in the background approach the magnitude of the plume enhancement. Cases exist in the literature where an aircraft enters an aged, very dilute plume, CO and acetonitrile rise, and $CO_2$ decreases. In this case (or in the less extreme case shown in Figure 1 of Yokelson et al (2013) the true original ER for $CO/CO_2$ cannot be retrieved regardless of whether the intercept is forced or not, or what type of regression is used. One would likely need a detailed history of all the air masses involved.

———————————————

---

## Author Comment (AC1) · 20 Feb 2018

Our responses and changes to the manuscript are detailed below. Referee comments are highlighted in bold text, with additions to the manuscript noted in plain text. In the revised manuscript, changes are also highlighted in red.

We note the jointly prepared comment by Clare Paton-Walsh and Bob Yokelson, which is an interesting and very useful addition. Given the conclusion of the comment is that no changes are required to the manuscript, a response is not necessary.

We have corrected some typographical errors in the manuscript as follows:

P5, L28: missing 'a' in 'interpreted as relative'.
P5, L28: 'degress' should be 'degrees'.
P12, L3. 'balanaced' should be 'balanced'.
P24, F1: 'Plume interception' in caption has an errant '1' on the end.

We have also added two additional studies to the comparisons and tables:

Stockwell, C. E., Jayarathne, T., Cochrane, M. A., Ryan, K. C., Putra, E. I., Saharjo, B. H., Nurhayati, A. D., Albar, I. Blake, D. R., Simpson, I. J., Stone, E. A., and Yokelson, R. J. (2016). Field measurements of trace gases and aerosols emitted by peat fires in Central Kalimantan, Indonesia, during the 2015 El Niño. Atmospheric Chemistry and Physics, 16(18), 11711–11732. http://doi.org/10.5194/acp-16-11711-2016

Desservettaz, M., Paton-Walsh, C., Griffith, David W. T., Kettlewell, G., Keywood, M. D., Vanderschoot, M. V., Ward, J., Mallet, M. D., Milic, A., Miljevic, B., Ristovski, Z. D., Howard, D., Edwards, G. C., and Atkinson, B. (2017). Emission factors of trace gases and particles from tropical savanna fires in Australia. Journal of Geophysical Research: Atmospheres, 122(11), 6059–6074. http://doi.org/10.1002/2016JD025925

**Referee #1: R. J. Yokelson comments**

We very much thank the reviewer for his detailed comments, which have significantly improved the manuscript.

**Several savanna fires and one forest fire were sampled in the near-field with a heavily instrumented aircraft in Brazil to measure emission factors (EF). These are the first EF measurements in Brazil since 2004 and 1996. The earlier studies sampled more fires, but there is way too little sampling of these extremely important, but variable sources. Further, this study used some instruments not previously deployed on fires in Brazil. The data should be published without a doubt. Since fires are variable, any EFs measured will fall on a Bell curve. Thus, the expectation is not that this data must agree with previous averages, but that it can be combined with previous data to nudge an evolving global average. At the same time, the authors find no evidence that erroneously low EF in previous work caused the ubiquitous low a-priori emissions in regional to global models.**

We thank the reviewer for his positive comments.

**There is one serious oversight that needs to be fixed. Ferek et al., (1998) reported EF as gC/KgC not g of species /Kg fuel. Assuming the fuel is about 50% C, their EF for BC need to be divided by two to compare to the work here. The EF for CO2 should be divided by two and then multiplied by (44/12) to convert to g CO2/ kg fuel, etc. The authors should recalculate all the Ferek et al numbers correctly and then update their comparisons, which come much closer in many cases if this step is taken.**

We thank the reviewer for highlighting this mistake in the manuscript. We note that Akagi et al. (2011) reported updated emission factors in their supplementary material for Ferek et al. (1998) based on correspondence with the authors and reanalysis of their published fire types. Given the revisions presented in Akagi et al. (2011), we have updated our figures for Ferek et al. (1998) to match.

Upon updating the values from Ferek et al. (1998), the major difference in our comparisons is for OC, which is now much closer to the Rondonia fire. This has been noted in the revised manuscript.

**Minor terminology point: "rainforest" is kind of colloquial. I suggest evergreen tropical forest throughout as distinct from seasonally dry tropical forest or just tropical forest if they are not sure which type. It is important to distinguish between understory and deforestation fires, which they do.**

We have replaced 'rainforest' with 'tropical forest' throughout the manuscript. This change includes the title of the manuscript, which is now 'Near-field emission profiling of Tropical Forest and Cerrado fires in Brazil during SAMBBA 2012'.

**One topic not mentioned that might make a good addition to the paper if possible. Were any aerosol optical properties measured? If so, did they scale with EC/OC or MCE as in Pokhrel et al., (2016)?**

Aerosol scattering and absorption were measured by a nephelometer and PSAP respectively. However the performance and time-resolution of the PSAP was not sufficient to calculate the single scattering albedo. We do not report scattering-related measurements as we prefer to focus on the chemical properties of the aerosol. Furthermore, measurements of aerosol optical properties will be a focus of a forthcoming manuscript detailing regional pollution during SAMBBA.

**P1, L12: I suggest defining organic aerosol as "(OA)". OA is the primary measurement, but OC is reported by dividing OA by 1.6. Reporting OC facilitates comparison with more historical data, but it might be worth reporting OA too?**

We have added the emission factor values for organic matter (OM) to the abstract so that the nomenclature is consistent with the rest of the manuscript.

**P1, L15: "fuel content" to "fuel mixture" maybe?**

Replaced.

**P1, L17: Perhaps change "scaling" to "scaling up" to set the context for why the possibility of low EF was interesting.**

Amended.

**P1, L18: Maybe simplify the end of this sentence as e.g. ". . . . one potential cause of low a-priori emissions in modeling studies.**

Left as is.

**P2, L12: delete "to"**

Deleted.

**P2, L15: There is a far more complex mix of burning in Brazil than just two fire types as discussed at length elsewhere (section 2.3, Yokelson et al., 2007). Referees complained about the length of that section, but maybe a broader summary is in order here. Something like ~"a range of climate and fire types occurs in Brazil and fire-impacted ecosystems include pure grassland, a gradient of wooded savannah into dry (seasonal) tropical forest (aka Cerrado), and evergreen tropical forest (Ward et al., 1992). In forested areas understory fires can occur, but deforestation fires to establish pastures or croplands, along with pasture maintenance and agricultural residue fires are the most common types of burning on an evolving heterogeneous landscape." You could cite our 2007 paper or any of the excellent papers cited there-in. Then mention (move here) the later statements (already with citations) about burning shifting from forests to savannas.**

We have added the following to the manuscript:

'A range of climate and fire types occurs in Brazil, with fire-impacted ecosystems including pure grassland, a gradient of wooded savannah into dry (seasonal) tropical forest and evergreen tropical forest (Ward et al., 1992, Yamasoe et al., 2000). Deforestation and Cerrado (savannah-like) fires are commonly used for land clearing and pasture maintenance (Martin et al., 2010), which leads to high levels of black carbon, organic matter and gas phase species in the atmosphere.'

**P2, L23: Actual landscape fires can rarely be characterized by a single stage, but instead there is a dynamic, variable mix of flaming and smoldering processes.**

We have added that fires are made up of 'a dynamic, variable mix of combustion phases'.

**P2, L29-30: By sampling more we get a better idea of the mean and range. We can also learn about the factors driving the variability, which I think this study may do thru its analysis.**

Noted.

**P3, L1: Insert "and other" before "regions" as the mix of AMS and SP2 been used in the US on chaparral fires (Akagi et al., 2012), prescribed forest fires (May et al., 2014), agricultural fires (Liu et al 2016), etc.**

We have added the additional references to the manuscript.

**P3, L1-4: The historical context and the contrast with the current effort is a bit oversimplified. Previous work was not necessarily less sensitive or unable to probe individual plumes. Yokelson et al., (2007) flew Artaxo's calibrated nephelometer to get real-time PM10 in numerous single plumes and computed PM10 EFs for Brazilian fires. Light scattering measurements are pretty sensitive. It could be that PM based on light scattering is not as accurate as an AMS, but evidently a nephelometer was used to scale the AMS data in this work. In SCAR-B in Brazil and SAFARI 2000 in Africa, a very large bag sampler was used and instantaneously filled in numerous individual plumes at multiple ages per plume. This allowed sensitive filter sampling, though filter artifacts can occur (e.g. Ferek et al., 1998; Sinha et al., 2003). Also, real-time data was acquired with a suite of PM instruments in individual plumes in the Mexican tropics in Yokelson et al., (2007, 2009, 2011). I think Capes et al., (2008) deployed an AMS on North African fires. To my knowledge this was the first study to use both AMS and SP2 in individual smoke plumes in the tropics. However, the sentence implying this is particularly true for airborne work should be deleted unless the authors can cite a ground-based study of individual tropical BB plumes that used both SP2 and AMS. Finally, depending on performance, different approaches have a different set of strengths and weaknesses.**

We appreciate the reviewer's comments on this section and have re-written the text to focus more on the ability to measure aerosol chemical composition, which we feel is the major instrumental advance presented in our manuscript given the known biases for filter-based sampling. We have also omitted the text in relation to low time resolution given the reviewer's note that a large bag sampler was used in previous measurements. We have also deleted the sentence regarding airborne and ground-based sampling.

The new discussion is as follows:

'Previous measurements of the chemical composition of particulate emissions from South American tropical biomass burning were conducted over a decade ago using filter-based sampling, which have known biases (e.g. Bond and Bergstrom, 2006; Chow et al., 2007; Lack et al., 2008; Petzold et al., 2013; Bond et al., 2013).'

**P4, L1-5: The authors estimated the mass of PM from scattering data to scale the AMS mass. They should report what mass scattering efficiency they used and an error estimate. Then a scaling factor of 2.69 +/- ? Further, if the SMPS could be used to measure the AMS collection efficiency, it seems like it could also be used for an independent estimate of the AMS scaling factor. On L5 add "forest fire" before "data".**

We calculated a mass scattering effiency of 5.98 m^2/g based on four biomass burning flights unaffected by the pinhole blockage. We will report this and the uncertainty in the scaling factor (2.69 ± 0. 3) in the updated manuscript.

We have updated the sentence in the manuscript, which now reads as:

'The applied scaling factor was 2.69 ± 0.3 based on measured mass scattering efficiencies of 16.1 ± 0.3 m2g−1 and 5.98 ± 0.3 m2g−1 for the partially blocked and unblocked flights respectively and is applied to the data for B737 in this study'.

We used the nephelometer rather than the SMPS or GRIMM data due to better instrument coverage during the campaign and to eliminate changes in the size distribution of the aerosol as a conflating factor in the comparison.

For the sake of comparison, we calculated a scaling factor of 3.02 using the GRIMM data.

**P4, L15: It's not super clear how coincidence was solved. Evidently they did not account for small rBC particles below the SP2 size cutoff as is sometimes done, but the figures seem to indicate this correction is not needed in this case.**

See response to referee #2 regarding the SP2 operation.

**P5, L15-20: Perhaps report both OA and OC and use OA along with the other species to estimate and compare fine PM?**

We have reported both organic matter (the AMS native measurement) and organic carbon.

**P5, L23: Technically "hydrocarbons" should be non-methane organic gases (NMOGs) since O-containing VOC dominate. Likewise on P6, L17**

Amended.

**P6, L2-3 don't need, smoke dilution rates slow down exponentially with time and can be even faster on ground-based measurements closer to the source.**

We have deleted 'The absolute concentration of trace gases and particulates in fire plumes cannot directly be used to interpret emissions due to the dilution of the species with the ambient background air. This is particularly important when sampling smoke from aircraft platforms.'

**P6, L22: A "few" percent is probably more accurate than 1-2% if particulate carbon, which was measured is not included in total C.**

Amended.

**P6, L26 "molecular" should be "atomic in the case of carbon (an error I have made many times).**

Amended.

**P7, L15: This is a nice description of the fire. Hotspots are shown in the figure on the day of sampling. It is stated there were no hotspots the previous day and the fire may have started the same day it was sampled. The fire seems a bit large to emerge in one morning and the authors could check a few of the previous days for hotspots since, due to cloud cover or orbital gaps any particular day may have no hotspot data. Brazilian fires can be anthropogenic and off-road due to the presence of indigenous peoples. We likely sampled some of these in an indigenous preserve along the Xingu River.**

We have clarified the following sentence 'MODIS hotspot data from the TERRA overpass at 14:26 UTC on 19 September 2012 indicated that this fire was likely started that day' in the

manuscript as we realise that it was unclear that we were stating that the fire likely started on the day prior to our flight (20 September 2012).

The revised sentence now reads as:

'MODIS hotspot data from the TERRA overpass at 14:26 UTC on 19 September 2012 indicated that this fire likely started on the day before our flight.'

We have added the following sentence to acknowledge that the fire may have been started by indigenous people:

'However, we cannot rule out that the fire may have been started by the presence of indigenous people, which would mean the fire was anthropogenic in origin.'

**P7, L26: 30 ppm is really fresh! That's a good sign that the EF are not distorted by mixing.**

Noted. No amendment made.

**P8, L24: The high r-squared likely indicates that the plume was well-mixed and the fire burned a homogeneous fuel bed as opposed to a "common source." (relevant also to P10, L8)**

Amended. Revised sentence:

'The trace gas species measured on the aircraft are very strongly correlated, with r-squared values between 0.92 and 0.99 illustrating that the plumes are well-mixed and that these active fires likely burned a homogeneous fuel bed.'

**P8, L25-26: Perhaps the authors are trying to say that there is more variability when sampling a group of fires than a single fire, but the group of fires has a fairly high r-squared nonetheless.**

This is correct. No amendment required.

**P9. L4-5: Also a general comment. The data for the Rondonia fire is in good agreement for PM and CH4 with the data for tropical dry forest fires in Mexico (Table 3 in Yokelson et al., (2011)). Yokelson, R. J., Burling, I. R., Urbanski, S. P., Atlas, E. L., Adachi, K., Buseck, P. R., Wiedinmyer, C., Akagi, S. K., Toohey, D. W., and Wold, C. E.: Trace gas and particle emissions from open biomass burning in Mexico, Atmos. Chem. Phys., 11, 6787-6808, doi:10.5194/acp-11-6787-2011, 2011.**

We thank the reviewer for highlighting this and note that the study is included in our comparison tables. Rather than comparing to every study detailed in our tables, we limited the comparisons to mainly studies in Brazil to improve the readability of the manuscript. No amendment required.

**P10, L2: It's worth clarifying if they are showing the size of the BC core or the core plus coating, and also confirming that coated BC particles were included in the computation of BC mass.**

We have noted that the size distributions refer to the BC core size. The instrumentation section states that the 'total mass of particles containing refractory black carbon' is determined i.e. all particles (both coated and uncoated) that contain BC are measured.

**P10, L15: I would delete "more than" and possibly add "approximately" in light of the uncertainties.**

Amended.

**P10, L25 – P11, L6: Several things can be quickly fixed in here and throughout. The Ferek et al values that are compared to need to be converted to g species per kg of fuel. After doing that, the Akagi et al review paper lists these values for Ferek et al as examples: Savanna (OC 2.94, BC 0.35 (note good agreement with Sinha et al., (2003) for African savannas)) and Tropical Forest (OC 4.34, BC 0.46). So for instance, on P11, L3, the Ferek et al value is 0.35 and not 0.7 for EFBC. The authors should go thru all the text, tables, and also figures if applicable, and update the values quoted for Ferek et al and the comparisons as well. Further, most of the "BC" measurements they compare to are actually "EC" measurements that can be impacted by charring of OC. (The PAX measurement in Stockwell et al are an exception.) This should probably be mentioned here (along with later mention) and might inform the comparison.**

See response to prior comment on this mistake.

**P11, L2: The Stockwell measurements were by in-situ photoacoustic spectroscopy. This may be a good place to mention EC artifacts and the possibility of lower EFBC because of an unusually smoldery fire. Nonetheless, the data can be factored into the global average nudging it down.**

We have added a reference to the Stockwell et al. (2016) results for black carbon and elemental carbon to the discussion:

'Stockwell et al. (2016) reported emission factors for black carbon and elemental carbon of $0.0055 \pm 0.0016$ g kg−1 and $0.24 \pm 0.10$ g kg−1 respectively, illustrating the significant differences in what is usually assumed to represent EFBC when using different measurement techniques.'

**P12, L22: change "to" to "as"**

Amended.

**P13, L9: change "is" to "was"**

Amended.

**P13, L26: Re "the instrument" – was the SAMBBA SP2 or the May et al SP2 or both SP2's calibrated with urban-BC relevant material? Suggest clarifying by changing to "our instrument" "both instruments", etc as appropriate.**

Amended. Both the May et al. and our SP2 were calibrated using urban-BC relevant material.

**P13, L27-28: Informational only, we have just completed such a comparison in FIREX**

Noted.

**P14, L19: If there is no clearing then a fire is not a deforestation fire, but understory fires can be indigenous/anthropogenic to promote favored tree species, for hunting, or to improve access.**

We have noted the following in the conclusion, which is supported earlier in our discussion of the fire in the results (section 3.1.1):

'We believe that the Rondonia fire was most likely a wildfire.'

**P14, L20: "numerous" may be a bit overstated for what looks like about ten hotspots in 3 groups. Maybe "several" or "a group of" is better.**

Amended.

**P14, L24: "illustrate" to "confirm" since the differences in initial emissions is a fairly well-known topic.**

Amended.

**Fig. 1. Nice fire pics, perhaps a larger version in supplement would be worthwhile?**

We will add the pictures as a supplement to the paper.

**Fig. 4. It seems odd that the values are larger for the Tocantins fires given the lower emissions??**

We're unsure what the referee is referring to here. The rBC number and mass values are indeed larger for the Tocantins fires given the rBC emissions were much greater than the Rondonia fire. Perhaps the referee has not seen the difference in the scales? With that in mind, we did note in the figure caption that the scales differ.

**Table 1: header, reference 9 is Akagi et al., (2012). "laboratory" entry could be "lab/field" since the cottonwood log was in lab, but the Zambian log was in the field.**

Amended.

**Table 2: There should not be any missing ("-") values since all three gases were measured in all cited studies. For the Brazil smoldering logs (ref 2) the value for CH4/CO (X1000) is 143 not 14.3. This value for ref 7 may also be a factor of ten low?**

We have added the missing values and corrected the typographical error for ref. 2.

**Table 3: Third and thirteenth entries down for EFCO2 (ref 3) looks suspicious.**

We have updated these values (for Ferek et al. (1998)) following the previous comments.

*Referee #2: G. R. McMeeking comments*

We thank the reviewer for their comments.

**The manuscript presents biomass burning emission measurements during a series of research flights over several fires in Brazil. This is a region of global significance in terms of both total particulate matter as well as black carbon emissions. While there have been several previous campaigns focused on characterizing emissions in this region, none have used more modern instruments, and the relative scarcity of data from this area coupled with its importance certainly merits publication in my opinion.**

**I have little to add beyond Dr. Yokelson's thorough comments, but do recommend addressing a couple of smaller issues/areas in more detail in the manuscript in a revised version:**

**It would be helpful to others in the SP2 community to know a bit more about how the instrument was operated in these more challenging conditions. Some small details regarding the sample flow rate and dilution (if any) could be provided. In addition, an estimate of the concentration limit where true particle coincidence (multiple BC present in the laser beam at the same time) would be helpful, and a verification that the field data remained below this value.**

We have noted the SP2 sampling conditions in the revised text as follows:

'The SP2 sample flowrate was approximately 120 vccm and operated without sample dilution.'

We have also added more details regarding the particle coincidence citing a subsequent analysis for very high direct diesel emissions using the Manchester aerosol chamber as described in Liu et al. (2017):

'An offline comparison of the SP2 with a Sunset OC/EC measurement at very high BC mass loadings was performed by measuring the direct diesel emissions using the Manchester aerosol chamber (Liu et al., 2017), which suggested a high correlation between both measurements under high BC mass loading (up to approximately 15 µgsm−3). Above this limit, the SP2 measurement was biased low. However, the BC masses in this study were well below this threshold, thus would not be affected by the coincidence issue.'

Liu, D., Whitehead, J., Alfarra, M. R., Reyes-Villegas, E., Spracklen, D. V., Reddington, C. L., … Allan, J. D. (2017). Black-carbon absorption enhancement in the atmosphere determined by particle mixing state. Nature Geoscience, 10(3). http://doi.org/10.1038/ngeo2901

**The manuscript mentions the use of dryers on the AMS and SP2 inlet line, but does not discuss losses. Deriving correction factors for losses in nafion dryers can be difficult, but some short discussion of potential impacts on measurement uncertainties would be useful. On a related point, I assume the nephelometer inlet line RH was at times quite different from the AMS/SP2 line, else the nephelometer could be used to apply corrections for the pin-hole issue earlier in the study at all times. I am curious if the pinhole blockage effects may have varied with sample line RH. Any systematic relationships between emission ratios to CO and RH might hint at this.**

We have added that nafion driers are subject to particle losses and that they represent an additional uncertainty, which we have not considered in our analysis as we assume these to be small compared to the instrument and variation in the ambient sampling.

We discovered an error in the text, which mistakenly stated that the pin-hole correction was applied to data where the sample line humidity was below 40%. We actually used all of the data during low-altitude segments as we did not observe a dependence on relative humidity. This has been corrected in the revised manuscript. We have also added that only slow-mode data from the AMS was used given the difficulty in matching 1Hz data from the AMS, SP2 and nephelometer data together.

**Suggest referring to BC "core" diameters rather than BC diameter to avoid potential confusion with the mixed particle size.**

Amended throughout.

**I recommend reporting an average OA mass concentration in Table 6. This can be useful for any future comparisons with EF measured at different concentrations and can help untangle potential impacts of semi-volatile partitioning.**

We note the comment but we have reported the range in OA mass concentration in Figure 2 and sections 3.1.1 and 3.1.2. We feel this is the most transparent method for describing the experimental conditions given the range in concentrations spanning two orders of magnitude, whereas a simple average would be misleading given this range.

No changes made to manuscript.

*Referee #3: C. Paton-Walsh comments*

We thank the reviewer for their comments.

**This is an excellent paper presenting some very valuable measurements of emissions from fires in Brazil, a poorly sampled region of the globe. The paper is definitely suitable for publication in ACP, and has been expertly reviewed already by Bob Yokelson and Gavin McMeeking. I have a few minor additional comments below:**

**1. It has become traditional (following Yokelson et al.,[1999]), when calculating emission ratios via the best straight-line fit to a plot of one species against the reference species, to first subtract the background amounts and then force the regression to go through zero. However, subtracting background amounts is not required, because this has no mathematical impact on the gradient of the best line fit. Forcing the line through zero may change the gradient, but it puts unnecessary weight to the background concentrations measured/assumed. If these are very close to the real (and unchanging) background amounts, then the change to the gradient that occurs when you force the line through zero will be small. If the background assumed is incorrect, or is changing, the effect can be quite significant, as pointed out in a later paper by Yokelson et al., [2013]. A generalised least squares regression (that takes into consideration the uncertainties in both x and y) is a**

**mathematically simpler and more accurate way to determine the emission ratio. I recommend this way to calculate emission ratios. It will not avoid all of the issues pointed out in Yokelson et al., [2013] if the background amounts are hugely variable, but it will minimize them compared to the calculation the authors have used in this study. Having said that, the high r-squared values lend confidence to the results in this study. If the authors are confident that they haven't biased their results significantly and do not wish to go back and recalculate the emission ratios, then I recommend that a sentence is added on this matter. The sentence should point out that forcing the regression through zero can bias the emission ratio if the background amounts assumed are wrong or change, but in this case they are confident they are not subject to the pitfalls described in Yokelson et al., [2013].**

We thank the reviewer for their comment and their previously referred to joint-comment with referee #1 on this issue, the conclusion of which is that no changes to the manuscript are necessary.

No changes made to manuscript.

**2. The use of the 1 sigma uncertainty of the best line fit as the total uncertainty in the emission ratio is not valid when the uncertainties in the individual points are correlated with one another (which they are in this case). Ideally you should undertake a proper uncertainty analysis of your measurements. As a minimum you should acknowledge that the uncertainties in each point are correlated and so your value of the uncertainties will be an underestimate (since it will include the random errors only).**

We have added the following to section 2.2.2:

'Uncertainties in the ERs are derived as the one standard deviation error in the slope of the line of best fit following e.g. Akagi et al. (2012). Such uncertainties will represent an underestimate as they only include random errors given that the uncertainties in each point are correlated.'

**3. Finally, I assume that the correction to the AMS data that was required as a result of the partial blockage of the inlet would have added to the measurement uncertainties? Again, if it is not feasible to undertake a proper uncertainty analysis, you should at least acknowledge this has not been done and mention the additional uncertainty in the text.**

This has been partially covered in the response to referee #1. We have added that this represents an additional uncertainty to our calculations.

---

## Author Comment (AC2) · 20 Feb 2018

[revised manuscript text omitted]